# MiNT: Multi-Network Transfer Benchmark for Temporal Graph Learning

**Kiarash Shamsi**[1]*   **Tran Gia Bao Ngo**[1]*   **Razieh Shirzadkhani**[2]*
**Shenyang Huang**[2,3,8]   **Farimah Poursafaei**[2,3]   **Poupak Azad**[1]   **Reihaneh Rabbany**[2,3,6]
**Baris Coskunuzer**[4]   **Guillaume Rabusseau**[2,5,6]   **Cuneyt Gurcan Akcora**[7]

[1]Department of Computer Science, University of Manitoba, [2]Mila - Quebec AI Institute,
[3]School of Computer Science, McGill University [4]University of Texas at Dallas,
[5]DIRO, Université de Montréal, [6]CIFAR AI Chair
[7]AI Initiative - University of Central Florida, [8]University of Oxford

## Abstract

Temporal Graph Learning (TGL) aims to discover patterns in evolving networks or temporal graphs and leverage these patterns to predict future interactions. However, most existing research focuses on learning from a single network in isolation, leaving the challenges of within-domain and cross-domain generalization largely unaddressed. In this study, we introduce a new benchmark of 84 real-world temporal transaction networks and propose **Temporal Multi-network Transfer (MiNT)**, a pre-training framework designed to capture transferable temporal dynamics across diverse networks. We train MiNT models on up to 64 transaction networks and evaluate their generalization ability on 20 held-out, unseen networks. Our results show that MiNT consistently outperforms individually trained models, revealing a strong relation between the number of pre-training networks and transfer performance. These findings highlight scaling trends in temporal graph learning and underscore the importance of network diversity in improving generalization. This work establishes the first large-scale benchmark for studying transferability in TGL and lays the groundwork for developing Temporal Graph Foundation Models. Our code is available at `https://github.com/benjaminnNgo/ScalingTGNs`

## 1   Introduction

Temporal graph learning has emerged as a vital area of research for modeling dynamic systems where relationships among entities evolve over time. Many real-world phenomena naturally form temporal graphs, including social interactions [31], blockchain transactions [22], biological networks [8], and communication systems [52]. Unlike static graphs, temporal graphs capture time-varying patterns, enabling more accurate forecasting, anomaly detection, and representation learning [34].

The recent success of large pre-trained models in natural language processing (NLP) [9, 10, 44] and computer vision (CV) [43, 6] has spurred interest in developing Graph Foundation Models [36]. These models use a *pre-train and transfer* strategy, where neural networks trained on large datasets can generalize to new tasks with minimal supervision [7, 54]. While this paradigm is well-established in NLP and CV, its application to graph data, especially temporal graphs, is still nascent.

Existing TGL literature typically focuses on training and evaluating models on a single temporal network [19, 46, 40]. These methods learn temporal patterns from a single graph's evolution, implicitly assuming that the temporal dynamics are unique and not generalizable. This practice

---

*Equal contribution

39th Conference on Neural Information Processing Systems (NeurIPS 2025) Track on Datasets and Benchmarks.

limits the potential to learn shared temporal structures across different networks and hinders transfer learning, especially in zero-shot scenarios where labeled data is unavailable for new graphs.

In this work, we challenge this limitation and ask two fundamental questions: (1) Can temporal graph models benefit from learning across multiple networks within a single domain? (2) Can these models generalize to previously unseen networks, including those from different domains?

To address these questions, we construct a benchmark of 84 temporal transaction networks derived from the Ethereum blockchain. These networks collectively contain over 3 million nodes and 19 million edges, reflecting real-world financial dynamics over multi-year periods. To study cross-domain generalization, we also incorporate eight temporal social interaction networks from online communities. This benchmark allows for systematic evaluation of scaling behavior, transfer learning, and zero-shot generalization in TGL.

We introduce **Temporal Multi-network Transfer (MiNT)**, the first algorithm to pre-train temporal graph neural networks (TGNNs) across multiple dynamic graphs. MiNT alternates between networks during training, resets historical embeddings to ensure independence, and uses network-agnostic validation for model selection. We train MiNT models on up to 64 transaction networks and evaluate them on held-out networks. Our results show that MiNT models outperform single-network models, with performance improving as the number of pre-training networks increases (see Figure 1). These findings reveal a neural scaling trend in TGL and highlight the potential for building generalizable temporal graph models.

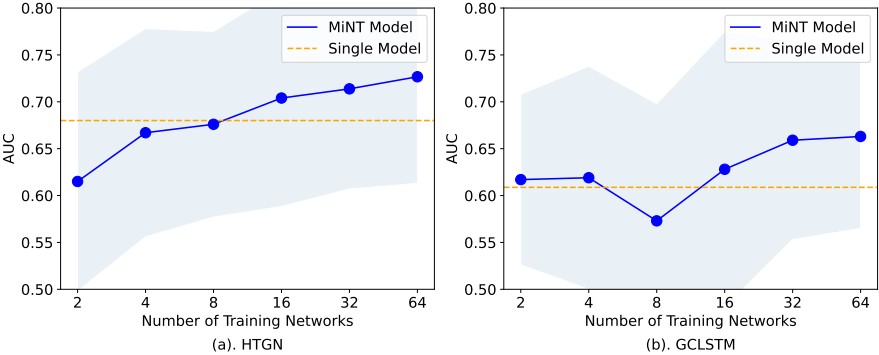

Figure 1: **Scaling behavior of MiNT on unseen networks.** Transferred inference performance of MiNT (multi-network model) on unseen networks, compared with standard training of individual networks (single model). The base TGNN models are (a) HTGN and (b) GC-LSTM. The metric is the average ROC AUC over 20 test networks.

**Our contributions are:**

- **Extensive Temporal Benchmark.** We release the first large-scale transfer learning benchmark of 84 Ethereum-based token networks and 8 existing social networks, enabling research on multi-network pre-training and generalization in TGL.
- **Multi-network Training Algorithm.** We introduce *MiNT-train*, the first algorithm to train TGNNs across multiple temporal graphs, leveraging order shuffling and context switching to ensure robustness and network independence.
- **Neural Scaling in TGL.** We empirically demonstrate that model performance improves with both the number of pre-training networks and the duration of training (in days), revealing a scaling trend analogous to those seen in NLP and CV foundation models for the first time.
- **Superior Zero-shot Transferability.** MiNT achieves competitive zero-shot performance on 20 unseen token networks, with the best average rank over all the baselines.

## 2   Related Work

**Temporal Graph Learning.**   Temporal graph neural networks have shown promising performance in tasks such as link prediction and node classification. Current literature [46, 62, 11] focuses on learning from a single network and partitioning the network into a training set and a test set chronologically. Thus, the objective is to extract patterns from the observed temporal graph and then

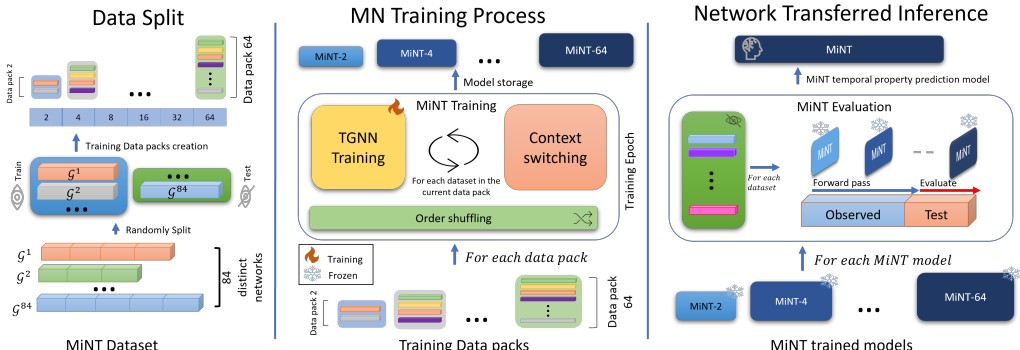

Figure 2: **MiNT framework.** Temporal graphs are preprocessed to generate discrete-time snapshots. The multi-network training pipeline leverages these snapshots to train TGNNs across multiple networks for network-transferred inference on unseen temporal networks. MiNT models of varying sizes (for example, MiNT-2, MiNT-4, MiNT-8, up to MiNT-64) each corresponding to a different number of training networks. Each model is trained on a distinct number of networks (e.g., 2, 4, 8, 16, 32, 64) and evaluated in a zero-shot setting on the same set of held-out test networks.

predict its future evolution. In the inductive settings, TGNNs predict accurately for novel nodes from the *same* network that were not observed in the training set [61, 56]. TGNNs have shown strong predictive performance but typically need large amounts of training data, which is often not available in practice. To mitigate this data scarcity, Agarwal et al. introduces a structured bipartite encoding mechanism that disentangles node representations from their features, enabling transfer of memory components from a source temporal graph to a target temporal graph on link prediction. In this work, we propose the first algorithm to effectively learn from multiple temporal graphs and focus on the temporal graph property prediction task. Additionally, we test the transferability of TGNNs on novel networks, unobserved during training.

Kazemi et al. [21] categorized temporal graphs into Discrete Time Dynamic Graphs (DTDGs) and Continuous Time Dynamic Graphs (CTDGs). In this work, we focus on DTDGs, since the temporal graph property prediction task is more appropriately defined over graph snapshots [50].

DTDG methods process each graph snapshot sequentially and use recurrent modules to learn temporal dependencies. For example, GCLSTM [11] stacks a graph CNN for feature extraction and an LSTM cell for temporal reasoning. In addition, leveraging the power of hyperbolic geometry, HTGN maps the temporal graph into hyperbolic space and utilizes hyperbolic graph neural networks and hyperbolic-gated recurrent neural networks to model the evolving dynamics. The SOTA for temporal graph property prediction is GraphPulse [50], which leverages topological data analysis to extract information from each snapshot. However, all DTDG models are designed to train and test on a single network, whereas this work explores multi-network training.

**Graph Foundation Models.** Recently, Xia and Huang [59] outlined the challenges associated with building a graph foundation model as structural heterogeneity, feature heterogeneity, fast adaptation, and achieving the neural scaling law. Neural scaling law is often used to categorize the relationship between model performance and factors in model training and design such as the number of parameters, the size of the training set and the amount of computation required [45, 20, 2]. Liu et al. [33] investigated neural scaling laws for static graphs by observing the performance of GNNs given the increase in the model size and training set size. Graph foundation models such as GraphAny [67] and AnyGraph [59] focus on designing architectures that allow for easy adaptation to unseen networks. There are also approaches to build domain-specific foundation models, such as those for molecular graphs [7, 53, 25]. In this work, we provide a novel collection of temporal graphs, which are necessary for building future TG foundation models. We also demonstrate the significant benefit of pre-training TGNNs on multiple networks for transferability to unseen networks.

**Zero-shot Inference.** Zero-shot learning has emerged as a powerful approach to enable pre-trained models to extrapolate predictions on unseen data from datasets used in the pre-training stage. Inspired by recent advancements in zero-shot learning and the power of pre-trained models in LLM [44] and CV [51],Wang et al. [54] introduced TEA-GLM, a novel framework that aligns GNN representations with LLM token embeddings by a linear projector. The incorporation of LLMs and GNNs enables

zero-shot inference on unseen graphs. Additionally, Xia et al. [60] proposed OpenGraph, an initiative promoting transparency, reproducibility, and community-driven advancements in graph representation learning. In this work, our proposed MiNT algorithm allows pre-trained TGNNs to achieve zero-shot inference without the need for fine-tuning or modifications, relying directly on the inference pass of a frozen pre-trained model.

**Benchmarks.** Due to space constraints, we defer a detailed discussion of graph benchmarks to Appendix B. In brief, while temporal graph benchmarks such as EdgeBank [42], GraphPulse [50], and TGB [19] have significantly advanced evaluation for within-network tasks, they remain limited for studying transferability, which requires training across a diverse set of distinct temporal networks.

## 3 Background

This section provides the background for temporal graph learning and the temporal graph property prediction task.

In this work, we focus on discrete-time dynamic graphs (DTDGs), which model temporal evolution through a sequence of graph snapshots. This choice aligns with the inherently discrete nature of our data source, the Ethereum blockchain, where thousands of transactions are batch-transferred in time-stamped blocks [49].

*Definition* 1 (**Discrete Time Dynamic Graphs**). DTDGs represent a network as a sequence of graph snapshots denoted as $\mathcal{G} = \{\mathcal{G}_1, \mathcal{G}_2, \mathcal{G}_3, \ldots, \mathcal{G}_n\}$. Each snapshot $\mathcal{G}_t = (\mathcal{V}_t, \mathcal{E}_t, \mathbf{X}_t, \mathbf{Y}_t)$ corresponds to the graph at timestamp $t$, where $\mathcal{V}_t$ and $\mathcal{E}_t$ are the sets of nodes and edges, respectively. $\mathbf{X}_t$ and $\mathbf{Y}_t$ are the node and edge feature matrices, respectively. A collection of DTDGs is defined as $D = \{\mathcal{G}^1, \mathcal{G}^2, \ldots, \mathcal{G}^m\}$, where $m$ is the number of DTDGs.

**Temporal Graph Property Prediction.** We consider a graph-level binary classification task where the goal is to predict whether a property of a discrete-time dynamic graph will increase or decrease over a future interval. Given a DTDG $\mathcal{G}$ and a current time $t_n$, we define a future window $[t_{n+\delta_1}, t_{n+\delta_2}]$ with $\delta_1 \leq \delta_2$. The model predicts the directional change of a chosen property, such as network growth:

**Definition (Network Growth).** Let $t_1$ and $t_n$ denote the start and end of the observation window, and $\delta_1, \delta_2$ define the prediction interval. Let $E(t_{n+\delta_1}, t_{n+\delta_2})$ be the multi-set of edges between times $t_{n+\delta_1}$ and $t_{n+\delta_2}$. The binary property $P$ is defined as:

$$P(\mathcal{G}, t_1, t_n, \delta_1, \delta_2) = \begin{cases} 1, & \text{if } |E(t_{n+\delta_1}, t_{n+\delta_2})| > |E(t_1, t_n)|, \\ 0, & \text{otherwise.} \end{cases} \tag{1}$$

This task enables reasoning about macro-level trends in dynamic networks. In social platforms, it can reveal shifts in user engagement, while in financial systems such as cryptocurrencies, it may signal changes in investor activity [4]. In addition to the network growth and shrinkage prediction task, MiNT is also evaluated on predicting the growth or shrinkage of the number of the largest connected component, which serves as an indicator of the structural coherence and robustness of temporal networks. Task definitions are provided in Appendix E.

## 4 MiNT:
## Temporal Multi-network Training

---

**Algorithm 1:** Multi-Network Transfer for Temporal Graphs

---

1: **Input:** A Temporal Graph Dataset $D = \{\mathcal{G}^1, \ldots, \mathcal{G}^m\}$, where $\mathcal{G}^i = \{\mathcal{G}^i_1, \ldots, \mathcal{G}^i_{n_i}\}$, $m$ = number of networks, $n_i$ is the number of snapshots of network $i$, a **TGNN** model, and a graph property prediction **Decoder**

2: **for** each *epoch* **do**
3:     Shuffle($D$) // *Order shuffling*
4:     **for** each $\mathcal{G}^i \in D$ **do**
5:         Initialize historical embeddings $\mathcal{H}_0$ // *Context switching*
6:         **for** $t = 1$ to $n_i$ **do**
7:             $\mathcal{H}_t = \mathbf{TGNN}(\mathcal{G}^i_t, \mathcal{H}_{t-1})$
8:             $\hat{y}_t = \mathbf{Decoder}(\mathcal{H}_t)$
9:             $\mathcal{L} = \mathbf{Loss}(y_t, \hat{y}_t)$
            Backpropagation
10:         **end for**
11:         Evaluate on validation snapshots of $\mathcal{G}^i$
12:     **end for**
13:     Average validation results across datasets
14:     Save the best model
15: **end for**

---

In this section, we introduce our Temporal Multi-network Training (MiNT) algorithm, an innovative multi-network pre-training framework designed to be applied to any backbone TGNN architecture for DTDG. By leveraging MiNT pre-training, the base TGNN model can transfer to unseen networks for inference. Figure 2 provides an overview of

our MiNT framework, illustrating the process from dataset curation to the model pre-training stage, and finally *network transferred inference* on test networks.

## 4.1 Multi-network Training

Existing temporal graph learning models typically train on a single temporal graph, limiting their ability to capture similar behaviors and generalize across different networks [46, 62]. In contrast, we consider a classical learning scenario where a training dataset of $m$ temporal graphs $D = \{\mathcal{G}^1, \mathcal{G}^2, \ldots, \mathcal{G}^m\}$ is drawn identically and independently (IID) from an unknown distribution, and the learned model is evaluated on a test set of unseen temporal networks.

Our MiNT algorithm trains across multiple temporal graphs by modifying a single network training model with two crucial steps: *shuffling* and *context switching*. As explained below, these steps render the algorithm network-agnostic, capable of learning from various temporal graphs to generalize effectively to unseen networks. Algorithm 1 shows the MiNT-train approach in detail. As the first step, we load the list of temporal graphs $D$, where each temporal graph $\mathcal{G}^i$ is represented as a sequence of snapshot $\{\mathcal{G}^i_1, \mathcal{G}^i_2, \ldots, \mathcal{G}^i_{n_i}\}$. For each epoch, we shuffle the orders of the list of datasets $D$ to preserve the IID assumption of neural network training.

**Order Shuffling.** Previous works focus on training models on a single network for temporal tasks; instead, we incorporate a shuffling step at each epoch to facilitate training on multiple networks and enable inference on unseen networks. The randomized ordering of networks during training at each epoch is important because it helps prevent the model from learning spurious correlations that could arise if the data were presented in a fixed order. Shuffling the datasets promotes randomness in the training process, contributing to more robust and generalizable model performance. Sequentially, we first initialize the historical embeddings, then train the model end-to-end on each dataset $\mathcal{G}^i$ in a similar manner to training a single model, and evaluate the performance on the corresponding validation set of dataset $\mathcal{G}^i$. After training on $m$ datasets from $D$, we compute the average validation results across these datasets. This average is used to select the best model, which is then used for inference. Early stopping is applied if needed. We verify the importance of order shuffling in the ablation study of Table 4.

**Context Switching.** Many TGNNs store and utilize node embeddings or latent states from previous timestamps at later timestamps; we refer to those embeddings as *historical embeddings* [62, 11, 40]. In Algorithm 1, this is represented in line 7 as

$$\mathcal{H}_t = \mathbf{TGNN}(\mathcal{G}^i_t, \mathcal{H}_{t-1}),$$

indicating that at time steps $t$, the temporal graph model takes as input both the current snapshot $\mathcal{G}^i_t$ and the *latent state* $\mathcal{H}_{t-1}$ from the previous time step (similar to RNNs). Resetting historical embeddings at the beginning of each epoch (line 5 of Algorithm 1) is a key step in training a temporal model across multiple networks for several reasons. First, it helps prevent the model from carrying over biases or assumptions from one network to another, ensuring that it can adapt effectively to the unique characteristics of each network. Second, it enables the model to learn the most relevant and up-to-date information from the current network, improving performance and generalization across different networks. This is equivalent to resetting the initial vector of recurrent neural networks at the beginning of each sequence.

**NTI: Network-Transferred Inference.** To evaluate the transferability of each multi-network model, we test the model on test sets that are unseen by the models during the training phase. We first divide our networks into two disjoint sets, where one set is used for training, obtained by randomly selecting 64 token networks, and the remaining 20 token networks are used to evaluate the performance. In the inference phase, we begin by loading all the weights of multi-network models, including the pre-trained encoder and decoder parameters, while initializing fresh historical embeddings. Then, we perform a single forward pass over the train and validation split to adapt the historical embeddings specific to the testing dataset.

## 4.2 MiNT Datasets

We construct a diverse collection of 84 large-scale ERC20 token networks derived from the Ethereum blockchain [58], capturing real transaction patterns from 2017 onward [5]. Each network reflects distinct investor behaviors and evolves independently, with varying start times and durations. Furthermore, edges have different types and scales per token. Hence, networks cannot be combined. These

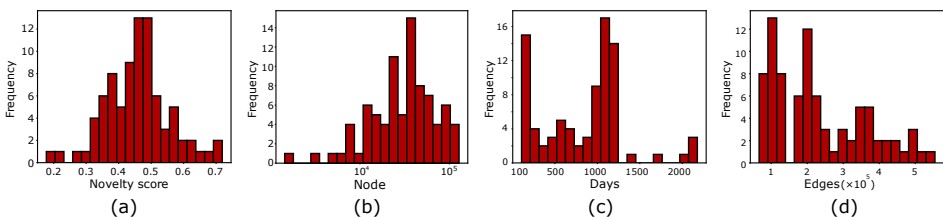

Figure 3: **Network statistics of MiNT networks.** (a) Novelty score, (b) number of nodes, (c) number of days, and (d) number of edges.

characteristics make the dataset well-suited for studying shared and unique temporal graph dynamics. The data extraction process is detailed in Appendix C.

Table 1 and Figure 3 summarize the diversity of the MiNT benchmark, illustrating variations in scale, temporal resolution, and structural dynamics across 84 token networks. Most networks contain over 10k nodes (investors) and 100k weighted directed edges (transactions), with lifespans ranging from 107 days to 6 years. Figure 3(a) reports novelty scores [42], showing that most networks exhibit daily novelty above 0.3, indicating a consistent influx of new edges. We follow a 70%-15%-15% split for train, validation, and test sets, and compute the surprise score [42] to assess edge uniqueness in the test set. As shown in Table 7, the networks have a high average surprise of 0.82, indicating that test edges are structurally dissimilar to those seen during training, making the prediction task more challenging. Appendix Figure 7 further compares the node, edge, and duration distributions of train and test sets, showing similar edge and time spans, with train sets typically containing more nodes.

We split the 84 networks into 64 for training and 20 for network-transferred inference. This partition allows robust evaluation of MiNT 's ability to generalize to entirely unseen temporal graphs. Additional statistics are provided in Appendix C.2.

Additionally, we adopt a diverse set of eight real-world social interaction networks. This evaluation aims to demonstrate that MiNT is not constrained to transactional graph domains and can effectively transfer learned representa-

Table 1: Summary of the 84 token networks in the MiNT benchmark.

| Category | Metric | Min | Max | Mean | Med. |
|---|---|---|---|---|---|
| **Scale** | #Nodes | 1.4K | 128K | 49K | 43K |
| | #Transactions | 78K | 555K | 312K | 298K |
| **Temporal** | Duration (days) | 100 | 2200 | 1080 | 960 |
| **Structural** | Novelty | 0.18 | 0.72 | 0.47 | 0.46 |
| | Surprise | 0.41 | 0.99 | 0.86 | 0.88 |

tions to structurally and semantically distinct networks. The selected social datasets are *LastFM*[48], *MathOverflow*[39], *SuperUser*[39], *Email-Eu*[39], *AskUbuntu*[39], *CollegeMsg*[38], *StackOverflow*[39], and *RedditB*[26]. These datasets span a wide range of social communication settings, from question-answering platforms to messaging and collaboration networks, providing a rigorous testbed for cross-domain transfer.

## 5 Experiments

We evaluate the transferability of our proposed MiNT approach on unseen test networks in both single-domain (Section 5.2) and cross-domain (Section 5.3) settings. Our code is available at `https://github.com/benjaminnNgo/ScalingTGNs`, and the datasets are hosted at `https://zenodo.org/records/15364297`. We begin by defining the baseline models and backbone architectures used in our experiments.

We focus on network growth or shrinkage [50] as the main prediction task. Additionally, we include the growth of the largest connected component to demonstrate the model's capability on a different property prediction task in Appendix I. Detailed task definitions are provided in Appendix E. As weekly forecasts are common in the financial context for facilitating financial decisions [23], we set $\delta_1 = 3$ and $\delta_2 = 10$ days for the tasks, thus predicting the temporal graph property over weekly snapshots.

In addition, to show that MiNT is agnostic to the underlying TGNN architecture, we select two widely-used TGNN models as our backbone: HTGN [62] and GCLSTM [11]. We formulate our datasets as discrete-time dynamic graphs, as our prediction tasks are defined over weekly graph snapshots that naturally capture financial and behavioral cycles observed in blockchain networks [1]. This formulation preserves the temporal evolution of transaction structures while maintaining consistency

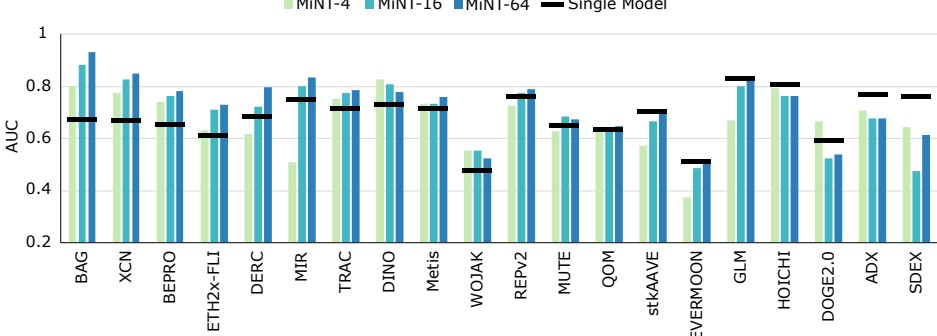

Figure 4: **MiNT Performance with Varying Training Scales.** Test ROC AUC of MiNT models trained on 4, 16 and 64 networks and evaluated on unseen test datasets. We compare the performance with HTGN single models trained and tested on each dataset.

with the aggregated nature of the data. While continuous-time dynamic graph models can also be applied to this domain, we limit the scope of this study to the discrete-time formulation, which is highly suitable for our prediction setting and data granularity. Based on this setting, we select HTGN, which has shown state-of-the-art performance on a similar task [50], and GC-LSTM, a robust baseline that effectively models temporal dependencies in discrete sequences. These two architectures provide strong and complementary backbones for evaluating the transferability and stability of MiNT. Experimental results suggest that HTGN has superior performance overall; thus, we adopt this model for in-depth analysis and ablation studies. [2]

## 5.1 Contenders And Baselines

For comparison, we include heuristics, single-network models, and our pre-trained MiNT models. We refer to models that are trained on only a single network (such as in existing literature) as *single models*. For each temporal graph, we use a $70\%$-$15\%$-$15\%$ split for training, validation, and test sets. During each epoch, the model processes snapshots sequentially in chronological order. Train and test networks share approximately 2% of their nodes (see Appendix Table 8). The results are reported with an average and standard deviation of three runs with different random seeds. We also provide our train/validation/test splits.

We train each model for 100 to 250 epochs with a learning rate of $1.5 \times 10^{-3}$, applying early stopping based on validation AUC with patience 20 and tolerance $5 \times 10^{-2}$. Binary Cross-Entropy Loss is used for evaluation, and Adam [24] for optimization. The graph pooling layer, loss function, and optimizer are shared in the multi-network training setup. The final pooling layer, implemented as a Multi-Layer Perceptron (MLP), computes the mean of node embeddings, concatenates it with four graph-level snapshot features (mean of in-degree, weighted in-degree, out-degree, and weighted out-degree), and outputs a binary prediction. The choice of mean pooling is further discussed in K, and Hyperparameter details are in Appendix F.

**Single Models.** We use five models from the literature, HTGN [62], GCLSTM [11], EvolveGCN [40], ROLAND [64], GraphPulse [50], WinGNN [68] and a naive heuristic baseline, Persistent Forecasting (P.F.), as our baseline models trained from scratch on individual networks. We explain each model in Appendix D.

**MiNT Models.** We train six multi-network transfer models, each with a different number of networks corresponding to $2^k$ datasets, where $k \in [1, 6]$. We name each multi-network model based on the number of datasets used in training; for example, MiNT-16 is trained with 16 datasets. The default TGNN architecture is HTGN, which shows superior performance, while the GCLSTM architecture is also trained and discussed in Table 2.

**Computational Resource.** We ran all experiments on NVIDIA Quadro RTX 8000 (48G memory) with 4 standard CPU nodes (either Milan Zen 3 2.8 GHz and 768GB of memory each or Rome Zen 2, 2.5GHz and 256GB of memory each).

---

[2]In what follows, the default MiNT models refer to HTGN with our MiNT pre-training, for example in Figure 4.

Table 2: **ROC AUC** scores of MiNT transfer models (with HTGN and GC-LSTM backbones) and single models (trained on test networks) on unseen test sets (average precision results and smaller MiNT results are reported in Appendix M). Best AUC in **bold**, second best underlined.

| Network | Single Model on Individual Networks | | | | | | | Transfer Model - 64 Networks | |
|---|---|---|---|---|---|---|---|---|---|
| | HTGN | GC-LSTM | EvolveGCN | GraphPulse | ROLAND | WinGNN | P.F. | GC-LSTM | HTGN |
| WOJAK | $0.479_{\pm0.005}$ | $0.484_{\pm0.000}$ | $0.505_{\pm0.023}$ | $0.467_{\pm0.030}$ | $\underline{0.529}_{\pm0.005}$ | $0.511_{\pm0.026}$ | 0.378 | $\mathbf{0.534}_{\pm0.020}$ | $0.524_{\pm0.027}$ |
| DOGE2.0 | $\mathbf{0.590}_{\pm0.059}$ | $0.538_{\pm0.000}$ | $0.551_{\pm0.022}$ | $0.384_{\pm0.180}$ | $0.513_{\pm0.022}$ | $0.577_{\pm0.038}$ | 0.250 | $0.551_{\pm0.022}$ | $0.538_{\pm0.038}$ |
| EVERMOON | $0.512_{\pm0.023}$ | $\mathbf{0.562}_{\pm0.179}$ | $0.451_{\pm0.046}$ | $0.519_{\pm0.130}$ | $0.349_{\pm0.119}$ | $\underline{0.525}_{\pm0.114}$ | 0.241 | $0.494_{\pm0.047}$ | $0.517_{\pm0.039}$ |
| QOM | $0.633_{\pm0.017}$ | $0.612_{\pm0.001}$ | $0.618_{\pm0.002}$ | $\mathbf{0.775}_{\pm0.011}$ | $0.641_{\pm0.003}$ | $0.645_{\pm0.099}$ | 0.334 | $0.618_{\pm0.004}$ | $\underline{0.647}_{\pm0.019}$ |
| SDEX | $\mathbf{0.762}_{\pm0.034}$ | $0.720_{\pm0.002}$ | $\underline{0.733}_{\pm0.028}$ | $0.436_{\pm0.030}$ | $0.483_{\pm0.254}$ | $0.726_{\pm0.000}$ | 0.423 | $0.723_{\pm0.002}$ | $0.614_{\pm0.020}$ |
| ETH2x-FLI | $0.610_{\pm0.059}$ | $0.670_{\pm0.009}$ | $0.688_{\pm0.010}$ | $0.666_{\pm0.047}$ | $0.621_{\pm0.023}$ | $0.617_{\pm0.056}$ | 0.355 | $\underline{0.697}_{\pm0.009}$ | $\mathbf{0.729}_{\pm0.015}$ |
| BEPRO | $0.655_{\pm0.038}$ | $0.632_{\pm0.019}$ | $0.610_{\pm0.012}$ | $\mathbf{0.783}_{\pm0.003}$ | $0.439_{\pm0.125}$ | $0.736_{\pm0.018}$ | 0.393 | $0.746_{\pm0.015}$ | $\underline{0.782}_{\pm0.003}$ |
| XCN | $0.668_{\pm0.099}$ | $0.306_{\pm0.092}$ | $0.512_{\pm0.067}$ | $\underline{0.821}_{\pm0.004}$ | $0.765_{\pm0.015}$ | $0.586_{\pm0.029}$ | 0.592 | $0.733_{\pm0.003}$ | $\mathbf{0.851}_{\pm0.043}$ |
| BAG | $0.673_{\pm0.227}$ | $0.196_{\pm0.179}$ | $0.329_{\pm0.040}$ | $\mathbf{0.934}_{\pm0.020}$ | $0.418_{\pm0.016}$ | $0.485_{\pm0.105}$ | 0.792 | $0.529_{\pm0.023}$ | $\underline{0.931}_{\pm0.028}$ |
| TRAC | $0.712_{\pm0.071}$ | $0.748_{\pm0.000}$ | $0.748_{\pm0.000}$ | $\underline{0.767}_{\pm0.001}$ | $0.495_{\pm0.223}$ | $0.752_{\pm0.007}$ | 0.400 | $0.742_{\pm0.004}$ | $\mathbf{0.785}_{\pm0.008}$ |
| DERC | $0.683_{\pm0.013}$ | $0.703_{\pm0.022}$ | $0.669_{\pm0.009}$ | $\underline{0.769}_{\pm0.040}$ | $0.405_{\pm0.357}$ | $0.674_{\pm0.044}$ | 0.353 | $0.696_{\pm0.011}$ | $\mathbf{0.798}_{\pm0.027}$ |
| Metis | $0.715_{\pm0.122}$ | $0.646_{\pm0.023}$ | $0.688_{\pm0.027}$ | $\mathbf{0.812}_{\pm0.011}$ | $0.696_{\pm0.108}$ | $0.690_{\pm0.039}$ | 0.423 | $0.697_{\pm0.013}$ | $\underline{0.760}_{\pm0.025}$ |
| REPv2 | $0.760_{\pm0.012}$ | $0.725_{\pm0.014}$ | $0.709_{\pm0.002}$ | $\mathbf{0.830}_{\pm0.001}$ | $0.751_{\pm0.003}$ | $0.744_{\pm0.026}$ | 0.321 | $0.733_{\pm0.019}$ | $\underline{0.789}_{\pm0.020}$ |
| DINO | $0.730_{\pm0.195}$ | $\mathbf{0.874}_{\pm0.028}$ | $\underline{0.868}_{\pm0.029}$ | $0.801_{\pm0.020}$ | $0.497_{\pm0.092}$ | $0.628_{\pm0.251}$ | 0.431 | $0.659_{\pm0.039}$ | $0.779_{\pm0.113}$ |
| HOICHI | $0.807_{\pm0.047}$ | $\mathbf{0.857}_{\pm0.000}$ | $\underline{0.856}_{\pm0.001}$ | $0.714_{\pm0.010}$ | $0.815_{\pm0.036}$ | $0.769_{\pm0.101}$ | 0.374 | $0.847_{\pm0.005}$ | $0.765_{\pm0.018}$ |
| MUTE | $0.649_{\pm0.015}$ | $0.593_{\pm0.030}$ | $0.617_{\pm0.010}$ | $\mathbf{0.779}_{\pm0.004}$ | $0.289_{\pm0.042}$ | $0.593_{\pm0.054}$ | 0.536 | $0.636_{\pm0.003}$ | $\underline{0.673}_{\pm0.013}$ |
| GLM | $\underline{0.830}_{\pm0.029}$ | $0.451_{\pm0.003}$ | $0.501_{\pm0.033}$ | $0.769_{\pm0.000}$ | $0.559_{\pm0.357}$ | $0.530_{\pm0.000}$ | 0.427 | $0.501_{\pm0.004}$ | $\mathbf{0.831}_{\pm0.024}$ |
| MIR | $0.750_{\pm0.005}$ | $0.768_{\pm0.026}$ | $0.745_{\pm0.015}$ | $0.689_{\pm0.097}$ | $0.228_{\pm0.060}$ | $0.742_{\pm0.015}$ | 0.327 | $\underline{0.788}_{\pm0.022}$ | $\mathbf{0.836}_{\pm0.016}$ |
| stkAAVE | $0.702_{\pm0.042}$ | $0.368_{\pm0.011}$ | $0.397_{\pm0.022}$ | $\mathbf{0.743}_{\pm0.006}$ | $0.591_{\pm0.122}$ | $0.572_{\pm0.018}$ | 0.426 | $0.650_{\pm0.028}$ | $\underline{0.709}_{\pm0.022}$ |
| ADX | $\underline{0.769}_{\pm0.018}$ | $0.723_{\pm0.002}$ | $0.718_{\pm0.004}$ | $\mathbf{0.784}_{\pm0.002}$ | $0.761_{\pm0.011}$ | $0.733_{\pm0.023}$ | 0.362 | $0.673_{\pm0.022}$ | $0.679_{\pm0.024}$ |
| Top rank ↑ | 2 | 3 | 0 | **8** | 0 | 0 | 0 | 1 | 6 |
| Avg. rank ↓ | 4.10 | 5.35 | 5.5 | 3.30 | 5.85 | 4.95 | 8.35 | 4.35 | **2.80** |

## 5.2 Single Domain Transfer Results

This section presents the network growth task performance of our multi-network models trained on the 64-token datasets and evaluated in zero-shot inference on 20 unseen token test datasets. Similar trends for the second task are reported in Appendix I. To evaluate that MiNT's transferability is not limited to the transaction domain, we also trained MiNT on social network datasets, where it demonstrates positive transfer. The detailed results are presented in Appendix H. We additionally investigate the (negligible) effect of data selection on the performance of MiNT, with detailed results provided in Appendix K.

Table 2 compares our results with the five baseline single models that are trained and tested on the same 20 datasets. We also report the top rank, average rank, and win ratio for each model. The top rank indicates the number of datasets where a method ranks first. To calculate the average rank, we assign an AUC-based rank (ranging from 1 to 9) to every model across the 20 test datasets and compute the average. The win ratio represents the proportion of datasets where a model outperforms a single model. Overall, MiNT-64 exhibits the best overall performance, achieving the state-of-the-art AUC performance in 6 networks and top two performance in 7 out of 20 total test networks with network-transferred inference.

Although the single GraphPulse model achieves the top rank of 8, it is a topology-based method without a GNN and requires supervised training on each dataset. In contrast, our GNN-based MiNT-64 model performs transferred inference efficiently across all datasets in just a few minutes. Despite some trained models like HTGN or GCLSTM excelling on specific datasets (e.g., SDEX and DINO), MiNT-64 consistently ranks competitively across the full benchmark, demonstrating strong generalization without per-dataset training.

For visual clarity, Figure 4 shows the AUC on test data results for MiNT-4, MiNT-16 and MiNT-64, as well as the single HTGN model. We show the performance of all six multi-network models in Appendix Figure 8. Overall, an upward trend is observed in most datasets from multi-network models 2 to 64, such as in *BAG*, *MIR*, and *BEPRO* datasets, highlighting the power of larger multi-network models in temporal graph learning.

Table 3: MiNT vs single model Performance Ranking.

| Model | Top rank ↑ | Avg. rank ↓ | Win ratio ↑ |
|---|---|---|---|
| Single model | 3 | 4.35 | – |
| MiNT-2 | 0 | 6.15 | 0.25 |
| MiNT-4 | 2 | 4.35 | 0.45 |
| MiNT-8 | 1 | 4.45 | 0.45 |
| MiNT-16 | 1 | 3.45 | 0.65 |
| MiNT-32 | 2 | 3.20 | 0.70 |
| MiNT-64 | **11** | **2.15** | **0.80** |

**Effect Of Scaling.** In Table 3, we further compare the models by reporting the top rank, average rank, and win ratio for different configurations of the multi-network models. We observe a notable improvement in performance as the number of training networks increases. For instance, the average

rank (lower is better) improves from 6.15 for MiNT-2 to 2.15 for MiNT-64, which signifies a roughly 50% performance enhancement when scaling from two networks to sixty-four. The improvement in the win ratio is also substantial, with MiNT-64 achieving the highest win ratio of 0.80, outperforming the other models in most datasets. This indicates that increasing the number of networks in a multi-network model significantly enhances its robustness and predictive power, particularly when compared to single models and smaller multi-network configurations. Overall, the multi-network-based models have shown superior zero-shot performance and transferability ability. We also conducted a study on the effect of the number of snapshots used for training, with the results presented in Appendix J.

**Ablation Study.**

We conducted an ablation study for the MiNT-train algorithm to assess the effects of *context switching* and *order shuffling*. Models are trained in the same way as a multi-network model training setup and tested on the 20 unseen test datasets. The average results are presented in Table 4. Training different multi-network models without resetting memory revealed that

Table 4: Ablation study results in ROC AUC.

| Model | MiNT-32 ↑ | MiNT-64 ↑ |
|---|---|---|
| Full Model | **0.714**± **0.107** | **0.727**± **0.114** |
| w/o Order shuffling | 0.708± 0.099 | 0.694± 0.109 |
| w/o Context Switching | 0.677± 0.098 | 0.688± 0.095 |

persistent memory across epochs negatively impacts generalization, emphasizing the importance of reset mechanisms to reduce overfitting. The gain from context switching is considerable when compared to the full model, as it enables stable zero-shot transfer across heterogeneous graphs. This mechanism prevents interference between structurally distinct networks during training and preserves embedding consistency across domains, which is essential for maintaining generalization under large-scale multi-network settings. Additionally, in Appendix K, we explored the necessity of shuffling data by fixing the order of training networks. The observed performance decline indicates that incorporating randomness to MiNT is vital for improving the model's robustness and generalizability.

## 5.3 Cross-domain Transfer Results

To further explore the potential of domain mixing, we train a combined model (*MiNT-12 Mix*) using six token networks and six social networks for the network growth task. The numbers are chosen to be equal to have a balanced mix. This transfer model is evaluated in a fully zero-shot setting across unseen networks from both domains. Results in Table 5 show that *MiNT-12 Mix* achieves the best average rank (1.91) and consistently places in the top two across 16 out of 20 datasets and outperforms *MiNT-12*, which is trained on 12 transaction networks. On social datasets, such as *RedditB* and *MathOverflow*, *MiNT-12 Mix* performs on par with, or better than, the standard HTGN, despite not being trained on any data from the evaluation set. On transaction networks such as *XCN* and *MIR*, it also outperforms both HTGN and MiNT models trained only on financial graphs. Table 5 shows the AUC

Table 5: **ROC AUC** scores of MiNT, HTGN and MiNT Mix. Scores in **first** and second. † indicates social networks.

| Network | Standard Training | Transfer Model | |
|---|---|---|---|
| | HTGN | MiNT-12 | MiNT-12 Mix |
| MIR | 0.750 ± 0.005 | 0.771 ± 0.038 | **0.779 ± 0.011** |
| DOGE2.0 | **0.590 ± 0.059** | 0.538 ± 0.000 | 0.538 ± 0.000 |
| MUTE | 0.649 ± 0.015 | **0.698 ± 0.033** | 0.660 ± 0.015 |
| EVERMOON | **0.512 ± 0.023** | 0.503 ± 0.037 | 0.438 ± 0.011 |
| DERC | 0.683 ± 0.013 | **0.722 ± 0.034** | 0.661 ± 0.006 |
| ADX | **0.769 ± 0.018** | 0.677 ± 0.014 | 0.712 ± 0.004 |
| HOICHI | 0.807 ± 0.047 | 0.795 ± 0.041 | **0.815 ± 0.012** |
| SDEX | **0.762 ± 0.034** | 0.425 ± 0.173 | 0.676 ± 0.050 |
| BAG | 0.673 ± 0.227 | **0.872 ± 0.029** | 0.838 ± 0.028 |
| XCN | 0.668 ± 0.099 | 0.761 ± 0.153 | **0.837 ± 0.014** |
| ETH2x-FLI | 0.610 ± 0.059 | **0.714 ± 0.014** | 0.670 ± 0.002 |
| stkAAVE | **0.702 ± 0.042** | 0.656 ± 0.010 | 0.615 ± 0.019 |
| GLM | **0.830 ± 0.029** | 0.811 ± 0.011 | 0.718 ± 0.045 |
| QOM | 0.633 ± 0.017 | **0.640 ± 0.011** | 0.631 ± 0.010 |
| WOJAK | 0.479 ± 0.005 | 0.521 ± 0.024 | **0.570 ± 0.033** |
| DINO | 0.730 ± 0.195 | **0.856 ± 0.035** | 0.846 ± 0.036 |
| Metis | 0.715 ± 0.122 | 0.697 ± 0.036 | **0.735 ± 0.024** |
| REPv2 | **0.760 ± 0.012** | 0.749 ± 0.017 | 0.756 ± 0.011 |
| TRAC | 0.712 ± 0.071 | 0.761 ± 0.034 | **0.768 ± 0.009** |
| BEPRO | 0.655 ± 0.038 | **0.765 ± 0.010** | 0.736 ± 0.041 |
| mathoverflow † | **0.788 ± 0.051** | 0.575 ± 0.195 | 0.782 ± 0.004 |
| RedditB † | 0.656 ± 0.040 | 0.653 ± 0.011 | **0.695 ± 0.004** |
| Top One ↑ | 8 | 7 | 7 |
| Top Two ↑ | 5 | 8 | 9 |
| Avg. Rank ↓ | 2.05 | 2.00 | 1.91 |

results of two Mix models compared to a single model on both social and transaction test networks.

These findings highlight the value of cross-domain pre-training: exposure to diverse structural and temporal patterns enables MiNT to develop representations that generalize effectively across domains.

The results support the broader hypothesis that multi-network pre-training can help build more robust temporal graph models.

**Time Complexity Analysis.** The MiNT-train algorithm has the same complexity as training the single model, but across all the training networks. Specifically, for the best performing HTGN-based model, the time complexity using MiNT-train is $O(m \cdot (N_{max}dd' + d'|\mathcal{E}_{max}|))$ where $m$ is the number of training networks, $N_{max}$ is set to the maximum number of nodes of networks in the training set, $d$ and $d'$ are the dimensions of the input and output features while $|\mathcal{E}_{max}|$ is the maximum number of edges in a snapshot. Empirically, we observe that the MiNT-training time scales linearly to the number of networks as seen in Figure 5, where we report the time per epoch for each multi-network model.

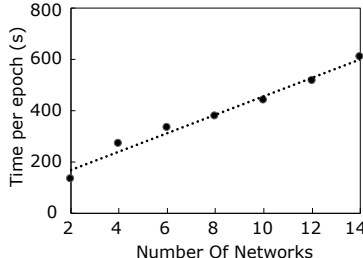

Figure 5: Time per epoch for training multi-network models.

A key strength of MiNT is its ability to perform zero-shot inference: once trained on multiple networks, it can generalize to unseen networks via a single forward pass without retraining. This makes MiNT highly efficient in real-world scenarios where new temporal graphs frequently emerge. To quantify this efficiency, we compare the time to train a single HTGN model on each test network with the inference time of a pretrained MiNT model, defining an *Efficiency Ratio* as the ratio between the two. As shown in Appendix Table 15, MiNT achieves over $180\times$ faster inference on average while maintaining competitive or superior performance. These results underscore the scalability and transferability of MiNT.

## Limitations & Future Work

Our work has the following limitations. i) Our scaling results show that training with a larger number of networks improves generalization. Although we limited the foundation model to sixty-four networks due to computational constraints, scaling to a broader set of graphs may further enhance transferability. ii) We focused on the DTDG setting, which effectively captures the temporal aggregation structure of our datasets. However, this choice restricts the current benchmark to discrete snapshots. Continuous-time formulations could also be applied to this domain, but would require dedicated event-level modeling and system design, which we consider an important direction for future work. iii) The current benchmark primarily includes transaction-based temporal graphs. Extending it with additional datasets from other domains, such as those in the Netzschleuder repository [41], could provide a more comprehensive evaluation environment and support broader assessment of model generalization across temporal graph types. iv) Beyond benchmarking, MiNT also opens several promising research avenues, including studying transferability mechanisms across dynamic systems, developing temporal pooling strategies for long-horizon reasoning, and exploring unified architectures toward temporal graph foundation models capable of cross-domain adaptation.

## 6 Conclusion

This work addresses a central question in temporal graph learning: can models trained on collections of temporal networks generalize to predict the evolution of previously unseen networks, both within and across domains? Our findings show that such generalization is not only feasible but effective.

We have introduced a benchmark of 84 Ethereum-based temporal graphs designed to support research on graph-level forecasting, neural scaling, and the development of foundation models for temporal graphs. To enable learning across diverse networks, we have proposed MiNT-train, the first framework for training temporal graph neural networks on multiple independent dynamic graphs.

Empirically, we have observed clear neural scaling behavior: model performance improves as the number of training networks and the number of snapshots increase. Additionally, MiNT-trained models achieve the highest average rank over single models on both within-domain and cross-domain test graphs. These results highlight the promise of multi-network training as a foundation for building generalizable temporal graph models and advancing the field toward temporal graph foundation models.

## Acknowledgments and Disclosure of Funding

This work was partially supported by the Canadian NSERC Discovery Grant RGPIN-2020-05665: Data Science on Blockchains, the National Science Foundation grants DMS-2202584 and DMS2220613, the Simons Foundation grant # 579977, and the Canadian Institute for Advanced Research (CIFAR AI chair program), the EPSRC Turing AI World-Leading Research Fellowship No. EP/X040062/1 and EPSRC AI Hub No. EP/Y028872/1. Shenyang Huang was supported by the Natural Sciences and Engineering Research Council of Canada (NSERC) Postgraduate Scholarship Doctoral (PGS D) Award and Fonds de recherche du Québec - Nature et Technologies (FRQNT) Doctoral Award. This research was also partially enabled by compute resources provided by Mila and UCF (mila.quebec, University of Central Florida).

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

# Appendix

## A    Broader Impacts

The proposed research on pre-training temporal graph neural networks across multiple networks has the potential to advance the field of machine learning and its applications significantly. By introducing methodologies to enhance the scalability and transferability of TGNNs, this work could revolutionize areas like network security, financial fraud detection, and real-time social network analysis, where dynamic and adaptive models are essential. The publicly available dataset of 84 Ethereum-based temporal networks will serve as a valuable resource for the research community, fostering innovation and collaboration. Furthermore, the principles of multi-network pre-training introduced here can inspire analogous advances in other temporal data domains, such as healthcare, transportation, and climate science. This research opens up a new direction in training generalizable temporal graph models that, for the first time, can be trained on distinct temporal networks, paving the way for Temporal Graph Foundation Models.

This work also introduces a set of Ethereum transaction token networks, which are publicly available to users who have the necessary resources, such as fast SSDs, large RAM, and ample disk space, to synchronize Ethereum clients and manually extract blocks. Additionally, all Ethereum data is accessible on numerous Ethereum explorer sites such as etherscan.io. An Ethereum user's privacy depends on whether personally identifiable information (PII) is associated with any of their blockchain address, which serves as account handles and are considered pseudonymous. If such PII were obtained from other sources, our datasets could potentially be used to link Ethereum addresses. However, real-life identities can only be discovered using IP tracking information, which we neither have nor share. Our data does not contain any PII. Furthermore, we have developed a request to exclude an address from the dataset.

## B    Extended Related Work on Graph Benchmarks

Benchmark datasets have become fundamental for advancing graph machine learning, providing a common ground to evaluate models and facilitate the development of graph foundation models. Early graph ML studies often relied on a handful of small, static benchmark graphs (e.g., citation networks like Cora/Citeseer and molecular graphs from the TU collection [37]). Repositories such as the Stanford SNAP dataset collection and the Network Repository cataloged many graphs for research use, but without standardized tasks or unified evaluation protocols [28, 47]. The lack of consistency in tasks and splits made it difficult to compare algorithms fairly. This motivated community efforts to create dedicated benchmark suites that are large-scale, diverse, and standardized.

### B.1    Static Graph Benchmarks

Static graph benchmarks focus on time-agnostic graph tasks (node classification, link prediction, graph classification) on fixed networks or sets of graphs. A seminal effort in this direction is the Open Graph Benchmark (OGB) [15], introduced at NeurIPS 2021. OGB provides a diverse collection of realistic graph datasets spanning domains such as social networks, citation networks, molecular graphs, knowledge graphs, and more. Importantly, OGB defines consistent evaluation protocols with meaningful train/validation/test splits (e.g., avoiding overly easy random splits) and unified metrics, addressing issues of reproducibility and out-of-distribution evaluation. The OGB suite includes challenging datasets like a citation network with hundreds of thousands of nodes (OGBN-Arxiv), a protein interface graph (OGBL-PPA) for link prediction, and molecule datasets for graph property prediction [15]. By benchmarking various GNN models on these datasets, Hu et al. showed that OGB tasks remain far from solved and identified key challenges such as scalability to large graphs and generalization under realistic splits. Following OGB's release, it has become a standard evaluation framework in graph ML research, with a public leaderboard tracking state-of-the-art results on each task.

Building on OGB, the community pushed toward even larger-scale benchmarks to catalyze foundation model-level advances. Hu et al. organized the OGB Large-Scale Challenge (OGB-LSC) in 2021, as part of the KDD Cup and later reported at NeurIPS 2021 [17]. OGB-LSC introduced three exceedingly large graph datasets, each targeting a core task at unprecedented scale: (1) a node classification task

on a heterogeneous academic graph with over 240 million nodes and 1.2 billion edges (MAG240M), (2) a link prediction task on a massive knowledge graph with 90 million entities (WikiKG90M), and (3) a graph regression task on a molecular dataset with $\tilde{4}$ million molecular graphs (PCQM4M) [17]. These datasets are orders of magnitude larger than prior benchmarks. The challenge drew 500+ teams, yielding innovative, scalable GNN approaches and significant performance improvements. Notably, the best-performing models in OGB-LSC employed techniques like deep transformer-based GNNs and aggressive neighbor sampling to handle scale (e.g., the Graphormer model, a Transformer tailored to graphs, won the molecular track) [63]. The OGB-LSC findings highlighted that expressive models can significantly outperform simpler, scalable baselines on large graphs, but also that many standard GNNs fail to even run at this scale without specialized training algorithms [17]. The annual OGB-LSC benchmark (continued in 2022) serves as a graph analog to ImageNet challenges, steering the community towards scalable graph learning techniques and pretraining strategies suited for extremely large data.

Another notable effort for static graphs is the Benchmarking Graph Neural Networks study by Dwivedi et al. [32]. Rather than introducing new data, the work systematically evaluated popular GNN architectures on a curated set of existing graph datasets (including both classical node/graph classification benchmarks and synthetic graph tasks). It revealed inconsistencies in prior evaluations and underscored the need for standard benchmarks like OGB. Overall, these static benchmark initiatives (OGB and others) have greatly improved the rigor and comparability of graph model evaluation. They also supply the data foundation for graph representation learning – for instance, OGB datasets are commonly used to pre-train GNNs or evaluate graph foundation models via fine-tuning.

## B.2 Temporal Graph Benchmarks

While static benchmarks assume a fixed graph structure, many real-world graphs are dynamic, evolving over time with new nodes or edges (e.g., social interactions, transaction networks, communication networks). Until recently, research on temporal graph neural networks relied on individually curated datasets and inconsistent protocols. For example, Kumar et al. (KDD 2019) introduced dynamic interaction graphs for Reddit and Wikipedia to evaluate embedding trajectory prediction [27], and various works used their own splits of social network data (e.g., user-item interaction logs, citation dynamics) to benchmark temporal GNN models. This fragmented landscape made it hard to gauge progress in learning on temporal graphs.

Recognizing this gap, Huang et al. proposed the Temporal Graph Benchmark (TGB), first released in 2023 [19]. TGB (accepted at NeurIPS 2023) is a unified collection of large-scale temporal graph datasets with standardized tasks and evaluation pipelines. It covers diverse domains, including social networks, communication, trade, finance, and transportation, reflecting the broad applicability of temporal graph learning. TGB defines two primary task categories: dynamic link prediction (predicting the future existence or properties of edges) and dynamic node property prediction (predicting future attributes or labels of nodes). Each dataset consists of a sequence of graph snapshots or time-stamped edge events spanning multiple years. For example, TGB includes a Wikipedia co-editing network (users editing pages over time), an E-commerce review network (Amazon user–product interactions over 20+ years), a Reddit reply network (users replying to each other over time), and an air traffic network (temporal flights between airports), among others. Crucially, TGB provides consistent train/validation/test splits along the timeline and an automatic evaluation pipeline, enabling reproducible benchmarking of temporal graph models.

In their extensive experiments, Huang et al. found that the performance of popular temporal GNN architectures varies wildly across different TGB datasets, and intriguingly, on certain dynamic node prediction tasks, simple time-series models can outperform complex temporal GNNs [19]. These insights point to open challenges and the need for better inductive biases in temporal models. TGB is an actively maintained project with a public leaderboard, poised to drive research in temporal graph representation learning. Recently, TGB 2.0 was introduced at NeurIPS 2024 to extend this benchmark to multi-relational dynamic graphs, i.e., temporal knowledge graphs and heterogeneous graphs with various edge types [13]. TGB 2.0 contributed eight new datasets spanning domains like social media, biomedicine, and communications, with up to tens of millions of time-stamped edges [13]. The inclusion of temporal knowledge graphs (e.g., evolving knowledge bases of events) bridges graph ML with temporal reasoning tasks from the knowledge graph community. Experiments in TGB

2.0 underscored that leveraging edge type (relation) information is crucial for high performance on multi-relational tasks, and again noted that simple heuristic methods can sometimes rival more sophisticated models [13]. However, most existing methods struggled to scale to the largest TGB 2.0 graphs, reinforcing the necessity for new scalable temporal GNN techniques. Together, TGB and its extension provide a comprehensive platform to evaluate how well algorithms handle the evolving, temporal aspect of graphs, complementing the static benchmarks like OGB.

## B.3 Blockchain Graph Benchmarks

One emerging application domain for graph learning is blockchain networks, which pose unique challenges and opportunities for benchmarks. Blockchains (like Bitcoin and Ethereum) can be represented as graphs (e.g., transaction networks where addresses are nodes and transactions are edges), often large-scale, dynamic, and multi-layered. For instance, the Bitcoin network continuously grows with each block of transactions, and Ethereum's smart contract calls form complex interaction graphs. Traditionally, machine learning on blockchain graphs was limited by data accessibility and labeling: researchers relied on isolated datasets curated for specific tasks, with no unified benchmark. A notable example is the Elliptic illicit transaction dataset [57], introduced in 2019, which consists of a Bitcoin transaction graph with ∼200K nodes labeled as licit or illicit. This dataset has been widely used to evaluate graph-based fraud detection and anti-money laundering models, and it established a baseline task of classifying illicit transactions on a large transaction graph. Other works have compiled datasets for tasks like Ethereum phishing address detection or DeFi fraud, but each dataset was used in a siloed manner in its respective paper [30].

To advance graph ML for blockchain data, Shamsi et al. introduced Chartalist [49], the first comprehensive repository of labeled blockchain graph datasets, which was published in the NeurIPS 2022 Datasets and Benchmarks track. Chartalist organizes blockchain data (from both UTXO-based blockchains like Bitcoin and account-based blockchains like Ethereum) into ML-ready graph datasets, complete with labels and task definitions. Importantly, it addresses the considerable preprocessing burden: raw blockchain ledgers are massive and require domain expertise to convert into meaningful graph features and labels. Chartalist provides cleaned and annotated graphs, including dynamic multi-layer networks extracted from blockchains. For example, it curates the evolving transaction graph of Bitcoin with ground-truth labels for certain known illicit events (like ransomware addresses), and similarly for Ethereum, it tracks address interaction networks with identified scams or anomalies [49]. By incorporating major blockchain events and annotating addresses (e.g., hacks, frauds), Chartalist enables supervised learning tasks such as address classification, transaction link prediction, temporal anomaly detection, and forecasting on blockchain graphs. This was a significant step, as previously no public benchmark existed for graph ML on blockchain data [49]. Chartalist's datasets are large-scale (the Bitcoin network graph in 2025 exceeds 400 million edges) and dynamic by nature, reflecting months or years of blockchain activity rather than static snapshots.

Recent benchmark efforts have further expanded blockchain graph datasets and tasks. One example is the "Multi-Chain Graphs of Graphs" dataset proposed by Luo et al. [35]. This work goes beyond single-chain analysis by constructing a hierarchical Graph-of-Graphs: local transaction graphs for multiple cryptocurrencies are connected via a higher-level graph that captures interactions between those cryptocurrencies (for instance, if one token is used to purchase another). Their dataset includes detailed labels at the token level and links across blockchains, supporting novel tasks like cross-chain link prediction and anomaly detection. This approach recognizes that modern blockchain ecosystems are interconnected (e.g., users swapping assets across chains), and analyzing them requires considering a network-of-networks structure. Another notable dataset is EX-Graph by Wang et al. [55], introduced at ICLR 2024, which bridges blockchain data with social networks. EX-Graph links the Ethereum transaction graph with the Twitter (X) social graph by identifying accounts that are active in both networks. It contains 2 million Ethereum addresses (nodes) and 30 million transaction edges, alongside 1 million Twitter user nodes and their following relationships, with over 30,000 address–social account linkages. By combining on-chain and off-chain (social) information, this benchmark allows researchers to study how incorporating external social features can improve blockchain analytics, for example, using Twitter interactions to predict cryptocurrency address behavior or to detect coordinated illicit activities. The introduction of EX-Graph underscores a trend of creating hybrid benchmarks that connect blockchain graphs with other data modalities to enrich learning signals.

It is worth noting that blockchain graphs also appear in the aforementioned broad benchmarks: for instance, the Temporal Graph Benchmark includes a cryptocurrency transaction dataset derived from Ethereum token transfers (a stablecoin transaction network during the 2022 Terra collapse) as one of its dynamic link prediction tasks [19, 49]. Similarly, TGB's dynamic node prediction tasks include a user–token interaction graph where the goal is to predict users' future activity with various cryptocurrencies [19]. The inclusion of these in TGB indicates a convergence where domain-specific efforts (like Chartalist) feed into general benchmark frameworks (like TGB). Going forward, we anticipate more blockchain-specific benchmarks to emerge, potentially covering areas like smart contract vulnerability graphs or transaction network simulations, given the growing interest in applying GNNs to cryptocurrency ecosystems. For now, Chartalist and its derivatives represent the state-of-the-art in providing public, labeled blockchain graph benchmarks for machine learning research.

## B.4 Benchmarks and Graph Foundation Models

The development of these benchmarks has been closely intertwined with progress on graph foundation models and training algorithms. By graph foundation models, we refer to large, general-purpose graph neural networks (or related architectures) that are trained on broad graph data (often via self-supervised learning) and can be adapted to a wide range of downstream tasks, analogous to NLP's pre-trained language models. High-quality benchmark datasets are a prerequisite for training and evaluating such models. For example, the massive node classification graph in OGB-LSC (MAG240M) and the huge molecular graph set in OGB have spurred research into pre-training GNNs on unlabeled graph data at scale [17]. Likewise, the diverse tasks in OGB (node, link, graph prediction across domains) provide natural downstream evaluations for a foundation model's versatility. We have started to see the emergence of self-supervised GNN training frameworks leveraging these benchmarks. Notably, Hu et al. proposed GPT-GNN [18], a generative pre-training method for GNNs using an attribute-masked graph generation task, which they demonstrated on the billion-edge Open Academic Graph (a subset of MAG) and an Amazon reviews graph. Their pre-trained model achieved significant gains on downstream node classification, showing the promise of foundation models on large graphs. Similarly, contrastive learning approaches like GraphCL [65] and graph autoencoders like GraphMAE [14] have been applied to OGB datasets to learn transferable representations. These algorithms create task-agnostic embeddings by maximizing agreement between differently perturbed versions of the same graph or by reconstructing masked features, enabling the model to capture generic graph structure and semantics.

Finally, benchmarks like TGB are driving advances in temporal graph learning algorithms that will feed into foundation models capable of handling dynamic data. The surprising observation that simple models sometimes beat complex temporal GNNs on TGB [19] suggests current architectures are not fully capturing temporal information; this has led researchers to rethink model designs (e.g., incorporating memory modules or temporal attention mechanisms) and training procedures for dynamic graphs. A foundation model that can jointly understand structure and temporal evolution might be trained by self-supervision on large temporal graphs (many of which are now available through TGB and related efforts). The multi-relational focus of TGB 2.0 [13] also pushes the development of models that can handle richly attributed graphs (multiple edge types, dynamic attributes), which is relevant for heterogeneous graph foundation models.

The ecosystem of graph and blockchain benchmarks, from static collections like OGB, to dynamic suites like TGB, and domain-specific data like Chartalist, provides the critical testbed and training ground for graph foundation models. These benchmarks cover a broad spectrum of graph scenarios that a foundation model should excel in: large-scale static networks, evolving temporal graphs, and complex multi-relational or cross-domain graphs (as in blockchains). By benchmarking new algorithms on these datasets, researchers can identify generalization gaps and scalability issues, guiding the design of more powerful graph neural network architectures. The continued expansion of benchmark datasets (especially at top venues like NeurIPS) ensures that, as graph ML enters the foundation model era, it does so on a firm, well-evaluated base.

**Why is the MiNT Benchmark Unique?** Our MiNT benchmark introduces a novel scale and structure for temporal graph learning by assembling 84 real-world ERC20 token transaction networks and 8 social interaction graphs, enabling both within- and cross-domain transfer studies. Unlike prior benchmarks such as TGB [19] and OGB [15, 16], which offer diverse but isolated dynamic

or static graph tasks, MiNT focuses specifically on the challenge of learning transferable temporal representations across independent dynamic graphs.

Each token network in MiNT is temporally disjoint, semantically distinct, and characterized by varying lifespan, novelty, and transactional behavior, making it unsuitable for naive aggregation or multi-label classification. This independence supports rigorous investigation of zero-shot generalization and pre-training on long-range temporal structures. Moreover, MiNT introduces network-level property prediction tasks, shifting the focus from local node- or edge-level tasks to macro-scale graph dynamics forecasting. It is the first benchmark to reveal neural scaling trends in temporal graph learning, demonstrating how increasing the number of training networks improves performance on unseen graphs. These properties position MiNT as the first foundation-model-oriented benchmark for temporal graph learning, complementing prior efforts by enabling systematic pre-training, ablation, and transfer evaluations across a controlled, large, and heterogeneous collection of temporal graphs.

Table 6: Comparison of temporal graph benchmarks.

| Dataset | # temporal graphs included | # newly introduced graphs |
|---|---|---|
| EdgeBank [42] | 13 | 6 |
| ROLAND [64] | 8 | 0 |
| TGB [19] | 9 | 8 |
| GraphPulse [50] | 9 | 7 |
| **Ours (MiNT)** | 92 | 84 |

## C  MiNT Datasets

Numerous graph benchmark datasets have been introduced to advance research within the temporal graph learning community. Poursafaei et al. [42] introduced six dynamic graph datasets while proposing visualization techniques and novel negative edge sampling strategies to facilitate link prediction tasks of dynamic graphs. Following the good practice from OGB [15], [19] introduced TGB, which provides automated and reproducible results with a novel standardized evaluation pipeline for both link and node property prediction tasks. However, these datasets belong to different domains, making them unsuitable for studying the scaling laws of neural network models trained with a large number of datasets from the same domain. [29] provides a temporal benchmark for evaluating graph neural networks in link prediction tasks, though their focus does not extend to training on multiple networks. Conversely, the Live Graph Lab dataset by [66] offers a temporal dataset and benchmark, employed for tasks like temporal node classification using TGNNs. This work aims to explore multi-network training and understand the transferability across temporal graphs. Therefore, we curate a collection of temporal graphs rather than focusing on individual ones as in prior work.

### C.1  Datasets Extraction

We utilize a dataset of temporal graphs sourced from the Ethereum blockchain [58]. In this section, we will describe Ethereum, explain our data pipeline, and conclude by defining the characteristics of the resulting dataset.

**Ethereum and ERC20 Token Networks.**  We create our transaction network data by first installing an Ethereum node and accessing the P2P network by using the Ethereum client Geth (`https://github.com/ethereum/go-ethereum`). Then, we use Ethereum-ETL (`https://github.com/blockchain-etl/ethereum-etl`) to parse all ERC20 tokens and extract asset transactions. We extracted more than sixty thousand ERC20 tokens from the entire history of the Ethereum blockchain. However, during the lifespans of most token networks, there are interim periods without any transactions. Additionally, a significant number of tokens live for only a short time span. To avoid training data quality challenges, we use 84 token networks with at least one transaction every day during their lifespan and are large enough to be used as a benchmark dataset for multi-network model training.

**Temporal Networks.** Each token network represents a distinct temporal graph, reflecting the time-stamped nature of its transactions. In these networks, nodes (addresses), edges (transactions), and edge weights (transaction values) evolve over time, capturing the dynamic behavior of the network. Additionally, these networks differ in their start dates and durations, introducing further variation in their evolution. While each token network operates independently with its own set of investors, they exhibit common patterns and behaviors characteristic of transaction networks. These similarities allow the model to learn and generalize from these patterns across different networks. Collecting temporal graphs from various ERC20 token networks allows for comparative analysis, uncovering

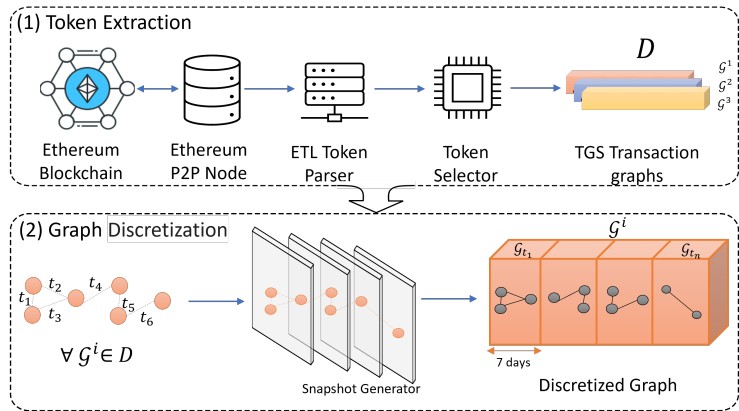

Figure 6: **MiNT data processing overview.** (1) *Token extraction*: extracting the token transaction network from the Ethereum node. (2) *Discretization*: creating weekly snapshots to form discrete time dynamic graphs.

common patterns and unique behaviors. This strengthens the model's ability to generalize and improves its robustness.

Figure 6 illustrates the MiNT overview from dataset extraction and discretizing graph networks for the model training step.

Table 7: All token networks' statistics.

| Token | #Node | #Transaction | Duration (days) | Growth rate | Novelty | Surprise | Token | #Node | #Transaction | Duration (days) | Growth rate | Novelty | Surprise |
|---|---|---|---|---|---|---|---|---|---|---|---|---|---|
| ARC | 11325 | 70968 | 606 | 0.43 | 0.32 | 0.88 | Metis | 52586 | 343141 | 907 | 0.44 | 0.48 | 0.89 |
| CELR | 65350 | 235807 | 1691 | 0.49 | 0.56 | 0.96 | cDAI | 52753 | 358050 | 1437 | 0.45 | 0.46 | 0.9 |
| CMT | 86895 | 205961 | 309 | 0.45 | 0.72 | 0.92 | BITCOIN | 34051 | 347054 | 178 | 0.48 | 0.39 | 0.63 |
| DRGN | 113453 | 341849 | 2164 | 0.44 | 0.57 | 0.97 | INJ | 60472 | 312822 | 1113 | 0.46 | 0.52 | 0.98 |
| GHST | 35156 | 180955 | 1146 | 0.43 | 0.51 | 0.93 | MIM | 23038 | 269366 | 885 | 0.44 | 0.4 | 0.89 |
| INU | 8556 | 66315 | 154 | 0.27 | 0.41 | 0.59 | GLM | 53385 | 234912 | 1080 | 0.5 | 0.53 | 0.96 |
| IOTX | 63079 | 288469 | 1993 | 0.45 | 0.56 | 0.99 | Mog | 14590 | 240680 | 107 | 0.37 | 0.38 | 0.55 |
| QSP | 117977 | 299671 | 2178 | 0.45 | 0.67 | 0.99 | DPI | 40627 | 234246 | 1150 | 0.49 | 0.5 | 0.86 |
| REP | 83282 | 224843 | 346 | 0.46 | 0.69 | 0.96 | LINA | 45342 | 227147 | 1144 | 0.45 | 0.46 | 0.95 |
| RFD | 23208 | 173695 | 169 | 0.3 | 0.39 | 0.6 | Yf-DAI | 22466 | 226875 | 1158 | 0.42 | 0.31 | 0.87 |
| TNT | 88247 | 316352 | 1216 | 0.43 | 0.55 | 0.93 | BOB | 42806 | 212099 | 199 | 0.35 | 0.48 | 0.73 |
| TRAC | 71667 | 299181 | 2110 | 0.46 | 0.54 | 0.97 | RGT | 35277 | 211932 | 1110 | 0.44 | 0.46 | 0.98 |
| RLB | 28033 | 240291 | 129 | 0.43 | 0.49 | 0.76 | TVK | 42539 | 208082 | 1062 | 0.41 | 0.48 | 0.93 |
| steCRV | 19079 | 211538 | 1033 | 0.45 | 0.53 | 0.9 | RSR | 50645 | 205906 | 659 | 0.47 | 0.62 | 0.91 |
| ALBT | 63042 | 434881 | 1152 | 0.43 | 0.44 | 0.89 | WOJAK | 34341 | 198653 | 201 | 0.37 | 0.48 | 0.73 |
| POLS | 128159 | 554705 | 1132 | 0.45 | 0.61 | 0.94 | ANT | 36517 | 200262 | 1107 | 0.47 | 0.46 | 0.93 |
| SWAP | 69230 | 509769 | 1213 | 0.46 | 0.45 | 0.79 | LADYS | 37486 | 192176 | 181 | 0.37 | 0.52 | 0.79 |
| SUPER | 83299 | 502030 | 986 | 0.47 | 0.46 | 0.85 | ETH2x-FLI | 11008 | 199088 | 965 | 0.47 | 0.28 | 0.84 |
| RARI | 87186 | 502960 | 1207 | 0.43 | 0.47 | 0.91 | TURBO | 38638 | 189048 | 189 | 0.33 | 0.48 | 0.72 |
| KP3R | 39323 | 493258 | 1102 | 0.43 | 0.33 | 0.88 | REPv2 | 39061 | 191367 | 1194 | 0.48 | 0.5 | 0.97 |
| MIR | 79984 | 444998 | 1066 | 0.45 | 0.43 | 0.92 | NOIA | 29798 | 185528 | 1133 | 0.46 | 0.37 | 0.7 |
| aUSDC | 23742 | 475680 | 1067 | 0.46 | 0.4 | 0.73 | 0x0 | 21531 | 182430 | 283 | 0.51 | 0.46 | 0.81 |
| LUSD | 25852 | 430473 | 943 | 0.48 | 0.36 | 0.87 | PSYOP | 25450 | 168896 | 169 | 0.32 | 0.39 | 0.59 |
| PICKLE | 28498 | 430262 | 1149 | 0.48 | 0.34 | 0.69 | ShibDoge | 40023 | 134697 | 680 | 0.43 | 0.53 | 0.8 |
| DODO | 47046 | 390443 | 1131 | 0.47 | 0.45 | 0.91 | ADX | 14567 | 123755 | 1188 | 0.44 | 0.4 | 0.91 |
| YFII | 43964 | 391984 | 1196 | 0.44 | 0.44 | 0.96 | BAG | 11860 | 122634 | 298 | 0.31 | 0.44 | 0.87 |
| STARL | 71590 | 369913 | 856 | 0.46 | 0.48 | 0.86 | QOM | 21757 | 118292 | 598 | 0.46 | 0.41 | 0.81 |
| LQTY | 34687 | 374230 | 943 | 0.45 | 0.34 | 0.91 | BEPRO | 26521 | 120261 | 1132 | 0.46 | 0.48 | 0.87 |
| FEG | 118294 | 367584 | 1007 | 0.4 | 0.62 | 0.92 | AIOZ | 29231 | 119926 | 947 | 0.43 | 0.49 | 0.89 |
| AUDIO | 91218 | 362685 | 1108 | 0.45 | 0.58 | 0.95 | PRE | 40476 | 118625 | 1113 | 0.5 | 0.55 | 0.86 |
| OHM | 45728 | 377068 | 690 | 0.43 | 0.46 | 0.88 | CRU | 19990 | 117712 | 1144 | 0.5 | 0.43 | 0.95 |
| WOOL | 16874 | 351178 | 716 | 0.41 | 0.18 | 0.41 | POOH | 27245 | 111641 | 193 | 0.26 | 0.49 | 0.69 |
| DERC | 24277 | 111205 | 824 | 0.45 | 0.49 | 0.83 | aDAI | 13648 | 187050 | 1068 | 0.45 | 0.46 | 0.82 |
| stkAAVE | 37355 | 110924 | 1128 | 0.42 | 0.57 | 0.71 | ORN | 44010 | 239451 | 1134 | 0.46 | 0.47 | 0.87 |
| BTRFLY | 8450 | 108371 | 453 | 0.48 | 0.34 | 0.44 | DOGE2.0 | 7664 | 79047 | 123 | 0.45 | 0.38 | 0.66 |
| SDEX | 9127 | 104869 | 240 | 0.41 | 0.44 | 0.75 | HOICHI | 5075 | 77361 | 436 | 0.36 | 0.32 | 0.71 |
| XCN | 20085 | 104185 | 607 | 0.46 | 0.42 | 0.84 | EVERMOON | 7552 | 79868 | 163 | 0.24 | 0.35 | 0.52 |
| HOP | 37004 | 102650 | 514 | 0.41 | 0.6 | 0.88 | MUTE | 12426 | 82345 | 977 | 0.43 | 0.46 | 0.95 |
| MAHA | 18401 | 96180 | 749 | 0.43 | 0.47 | 0.91 | crvUSD | 2950 | 88647 | 174 | 0.61 | 0.37 | 0.73 |
| DINO | 15837 | 94140 | 358 | 0.44 | 0.44 | 0.74 | SLP | 6675 | 95368 | 1151 | 0.43 | 0.36 | 0.91 |
| bendWETH | 1454 | 96898 | 593 | 0.51 | 0.21 | 0.51 | sILV2 | 12838 | 92905 | 611 | 0.4 | 0.34 | 0.48 |
| PUSH | 14501 | 93103 | 936 | 0.46 | 0.38 | 0.83 | SPONGE | 25852 | 90468 | 184 | 0.31 | 0.66 | 0.81 |

## C.2 Dataset Statistics

Our MiNT dataset is a collection of $84$ ERC20 token networks derived from Ethereum from 2017 to 2023. Each token network is represented as a dynamic graph, in which each address and transaction between addresses are a node and a directed edge, respectively. The biggest MiNT token network contains $128,159$ unique addresses and $554,705$ transactions, while the smallest token network has $1,454$ nodes.

Figure 3 shows that most networks have more than 10k nodes and over 100k edges. The lifespan of MiNT networks varies from $107$ days to 6 years, and there exists at least one transaction each

day. Figure 3.a shows the novelty scores, i.e., the average ratio of unseen edges in each timestamp, introduced by [42]. Figure 3 shows that most of the $84$ networks have novelty scores greater than $0.3$, indicating that each day sees a considerable proportion of new edges in these token networks. We adopt a $70\% - 15\% - 15\%$ split of train-test-validation for each token network and calculate the surprise score [42], which indicates the number of edges that appear only in the test data. As Table 7 shows, the token networks have quite high surprise values with an average of $0.82$. We also provide the node, edge, and length distribution for train and test sets separately in Figure 7. Overall, train set datasets mostly have more nodes than those in the test set, while the number of edges and days are in the same range for both.

We summarize detailed statistics of each token network in MiNT datasets in Table 7. In the table, the growth rate is the ratio of label $1$, indicating the increase in the number of edge counts concerning the problem definition defined in Appendix section E. In addition, we use the novelty and surprise scores introduced by Poursafaei et al. [42]. The novelty score is defined as the average ratio of new edges in each timestamp. The surprise score is defined as the ratio of edges that only appear in the test set. Formally,

$$novelty = \frac{1}{T} \sum_{t=1}^{T} \frac{|E^t \setminus E^t_{seen}|}{|E^t|}, \tag{2a}$$

$$surprise = \frac{|E_{test} \setminus E_{train}|}{|E_{test}|}, \tag{2g}$$

where $E^t$ and $E^t_{seen}$ denotes the set of edges present only in timestamp $t$ and seen in previous timestamps, respectively. $E_{test}$ represents edges that appear in the test set, and edges appearing in the train set are represented as $E_{train}$.

**Comparison Between Training And Testing Set**. Nodes, transactions, and length (in days) distribution over the training and testing sets are shown in Figure 7. Training sets well-support the multi-network model to generalize characteristics of the entire MiNT dataset due to the similarity between nodes, edge and length in days distributions shown in Figures 7a, 7b, 7c and those distributions across $84$ token networks of MiNT datasets. In addition, the variance of datasets' characteristics of the testing set is shown in Figures 7d, 7e and 7f.

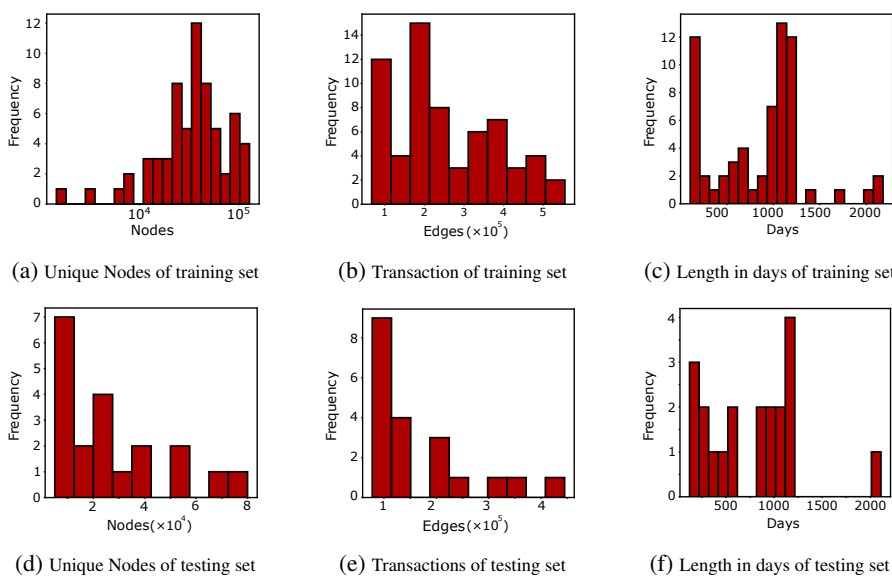

(a) Unique Nodes of training set    (b) Transaction of training set    (c) Length in days of training set

(d) Unique Nodes of testing set    (e) Transactions of testing set    (f) Length in days of testing set

Figure 7: Distribution of the characteristics of the datasets over training and testing sets.

**Node Overlap Analysis.** We analyze the overlap of nodes between different datasets and within each dataset, which helps demonstrate the highly dynamic nature of our datasets. Specifically, we compared the nodes in each test network with those in the training networks and calculated the average

overlap. As shown in Table 8, on average, only 2% of the nodes are common between the training and test datasets, highlighting the rapidly changing structure of these networks. Furthermore, we analyzed the node overlap within each test dataset by splitting it into the standard train-validation-test setup. We compared the nodes in the 70% training snapshots with the nodes in the final 15% test snapshots, and on average, only 4% of the nodes overlapped. This indicates the highly inductive nature of our model and emphasizes the zero-shot challenge it addresses in this domain. These findings underscore the importance of tackling such dynamic and evolving challenges in temporal graph learning.

# D   Temporal Graph Learning

In this section, we give further details about the temporal graph learning models we used as a baseline for our work.

**Persistence Forecast (P.F)** uses data from the previous and current weeks to predict the next week's property. If we observe an increasing trend in the number of transactions in the current week compared to the previous week, we predict a similar increasing trend for the following week. This simple model is based on the assumption that trends in transaction networks can persist over time. Our baseline method has three key aspects. First, we do not use any future information to generate the labels. Second, we compare the current week's transaction count to that of the previous week to determine the trend. Finally, if the current week shows an increase, we predict the same trend for the next week. This straightforward approach provides a basic baseline for comparison against more sophisticated predictive models.

Table 8: Overlapping Nodes Statistics

| Dataset | Avg. Common Nodes vs Train Set (MiNT-64) | Train vs Test Node Overlap |
|---|---|---|
| MIR | 0.021 ± 0.019 | 0.007 |
| DOGE2.0 | 0.026 ± 0.033 | 0.015 |
| MUTE | 0.033 ± 0.020 | 0.045 |
| EVERMOON | 0.023 ± 0.033 | 0.043 |
| DERC | 0.020 ± 0.020 | 0.031 |
| ADX | 0.024 ± 0.020 | 0.018 |
| HOICHI | 0.023 ± 0.013 | 0.053 |
| SDEX | 0.024 ± 0.019 | 0.141 |
| BAG | 0.019 ± 0.017 | 0.107 |
| XCN | 0.016 ± 0.010 | 0.034 |
| ETH2x-FLI | 0.038 ± 0.041 | 0.028 |
| stkAAVE | 0.026 ± 0.027 | 0.057 |
| GLM | 0.014 ± 0.015 | 0.047 |
| QOM | 0.018 ± 0.014 | 0.044 |
| WOJAK | 0.025 ± 0.032 | 0.018 |
| DINO | 0.018 ± 0.014 | 0.049 |
| Metis | 0.020 ± 0.013 | 0.041 |
| REPv2 | 0.016 ± 0.017 | 0.013 |
| TRAC | 0.015 ± 0.016 | 0.031 |
| BEPRO | 0.023 ± 0.022 | 0.021 |

**HTGN** leverages the power of hyperbolic geometry, which is well-suited for capturing hierarchical structures and complex relationships in temporal networks. HTGN maps the temporal graph into hyperbolic space and utilizes hyperbolic graph neural networks and hyperbolic gated recurrent neural networks to model the evolving dynamics. It incorporates two key modules that are hyperbolic temporal contextual self-attention (HTA) and hyperbolic temporal consistency (HTC)-to ensure that temporal dependencies are effectively captured and that the model is both stable and generalizable across various tasks [62].

**GraphPulse** addresses the challenge of learning from nodes and edges with different timestamps, which many existing models struggle with. It combines two key techniques: the Mapper method from topological data analysis to extract clustering information from graph nodes and Recurrent Neural Networks (RNNs) for temporal reasoning. This principled approach helps capture both the structure and dynamics of evolving graphs [50].

**GCLSTM** combines a Graph Convolutional Network (GCN) and Long Short-Term Memory (LSTM) units to handle both the structural and temporal aspects of evolving networks. The GCN is used to capture the local structural properties of the network at each snapshot, while the LSTM learns the temporal evolution of these snapshots over time [11].

**EvolveGCN** is designed to capture the temporal dynamics of graph-structured data. Instead of relying on static node embeddings, EvolveGCN evolves the parameters of a graph convolutional network (GCN) over time. By using a recurrent neural network (RNN) to adapt the GCN parameters, this model is capable of dynamically adjusting during both training and testing, allowing it to handle evolving graphs, even when node sets vary significantly across different time steps [40].

**ROLAND** is a dynamic graph learning framework that models node representations as hierarchical states, updated recurrently to capture temporal dependencies in evolving graphs. It supports scalable training using techniques like truncated backpropagation through time and meta-learning. In our DTDG setting, we use ROLAND to benchmark its performance and adaptability across diverse temporal networks [64].

**WinGNN** employs a lightweight GNN to capture the graph's structural features, similar to prior approaches. To address temporal dependencies, it introduces a unique random gradient aggregation mechanism combined with meta-learning. Specifically, WinGNN computes snapshot-level losses and propagates their gradients forward to model temporal evolution without relying on recurrent units. A randomized sliding window is further applied to extract window-aware gradients across snapshots, which are then aggregated to update the GNN parameters effectively [68].

## E  Temporal Graph Property Prediction

We define graph property prediction as the task of forecasting a specific structural property of a temporal graph over a future time interval. In this work, we focus on two binary classification tasks: predicting the growth or shrinkage of (i) transaction volume (i.e., edge count), and (ii) the size of the largest connected component (LCC).

In the network growth prediction task, the goal is to determine whether the number of transactions will increase in the upcoming time window relative to a preceding interval. Given the current weekly snapshot of a network, the model predicts whether transaction activity will rise or decline in the following week. This task is particularly relevant in financial domains, where fluctuations in transaction volume can reflect shifts in user engagement, liquidity, or investor interest. We adopt the same evaluation setup used in GraphPulse [50], and define the property formally as follows:

**Definition (Network Growth).** Let $t_1$ and $t_n$ denote the start and end of the observation window, and $\delta_1, \delta_2$ define the prediction interval. Let $E(t_{n+\delta_1}, t_{n+\delta_2})$ be the multi-set of edges between times $t_{n+\delta_1}$ and $t_{n+\delta_2}$. The binary property $P$ is defined as:

$$P(\mathcal{G}, t_1, t_n, \delta_1, \delta_2) = \begin{cases} 1, & \text{if } |E(t_{n+\delta_1}, t_{n+\delta_2})| > |E(t_1, t_n)|, \\ 0, & \text{otherwise.} \end{cases} \tag{3}$$

**Why is this task useful?** The network growth/shrink property prediction in financial networks forecasts changes in transaction numbers (edge count), revealing trends in investment activity. A growth in edge count indicates increased investor engagement, while a shrinkage suggests reduced activity or market hesitation. Such investor behavior impacts token prices, and analyzing the behavior helps guide investment strategies, resource allocation, and risk management by providing insights into the evolving dynamics of token networks. For social networks, network growth in time requires resource (e.g., server) allocation and maintaining dynamic load balancing. As a result, forecasting the growth allows for efficient planning.

**Definition (LCC Growth).** The second prediction task focuses on structural connectivity. Let $C_t$ denote the size of the largest connected component (LCC) at time $t$. The model predicts whether the LCC will grow over the upcoming interval. Formally:

$$P(\mathcal{G}, t_1, t_n, \delta_1) = \begin{cases} 1, & \text{if } |C(t_{n+\delta_1}, t_{n+\delta_2})| > |C(t_1, t_n)|, \\ 0, & \text{otherwise.} \end{cases} \tag{4}$$

**Why is this task useful?** In Ethereum token networks, the growth of the largest connected component reflects increasing structural integration, where more addresses become part of a unified transaction graph. This is important because token networks typically evolve through isolated pairwise trades, leading to many disconnected components or "islands" of investors. Such fragmentation limits information flow and liquidity, which can undermine price stability and market efficiency. A growing LCC, by contrast, indicates expanding interaction and network cohesion, which are often associated with higher liquidity, stronger network effects, and sustained adoption. Predicting LCC growth helps identify tokens that are moving toward broader market integration.

Setting $n = 7$, $\delta_1 = 3$, and $\delta_2 = 10$ days, we establish a practical graph property with a 7-day prediction window. This choice is particularly relevant in financial contexts, such as Ethereum asset networks, where it can guide investment decisions, and in social network infrastructure, like Reddit, where it supports maintenance planning.

While this work focuses on specific properties, numerous other characteristics, such as temporal triangle counting that can identify wash trades [12], can also be defined in this domain to highlight the significance of temporal graph property predictions.

## F  Hyperparameters

**Single Models.** We adopt a $70\% - 15\% - 15\%$ split ratio for the train, validation, and test, respectively, for each token network, and during each epoch, the training model processes all snapshots in chronological order. We train every single model for a minimum of 100 and a maximum of 250 epochs with a learning rate set to $15 \times 10^{-4}$. We apply early stopping based on the AUC results on the validation set, with patience and tolerance set to 20 and $5 \times 10^{-2}$, respectively. Specifically, in HTGN training, the node embeddings are reset at the end of every epoch. We use Binary Cross-Entropy Loss for performance measurement and Adam [24] as the optimization algorithm. It is important to note that the graph pooling layer, performance measurement, and optimization algorithm are also shared by the multi-network model training setup.

**Multi-network Models.** While following a similar training approach as in the single model training, we make specific adjustments for the multi-network model training. We set the number of epochs to 300 with a learning rate of $10^{-4}$ and a train-validation-test chronological split ratio same as single models. Early stopping is applied based on the validation loss with a tolerance of $5 \times 10^{-2}$ and the patience is set to 30. The best model is selected based on the validation AUC and used to predict the unseen test dataset.

## G  Hyperbolic Temporal Graph Network

Hyperbolic geometry has been increasingly recognized for its ability to achieve state-of-the-art performance in several static graph embedding tasks [62]. HTGN is a recent hyperbolic work that shows strong performance in learning over dynamic graphs in a DTDG manner. The model employs a hyperbolic graph neural network (HGNN) to learn the topological dependencies of the nodes and a hyperbolic-gated recurrent unit (HGRU) to capture the temporal dependencies. Temporal contextual attention (HTA) is also used to prevent recurrent neural networks from only emphasizing the most nearby time and to ensure stability, along with generalization of the embedding. In addition, HTGN enables updating the model's state at test time to incorporate new information, which makes it a good candidate for learning the scaling law of TGNNs. In our MiNT framework, we use the HTGN architecture as part of our multi-network model because it excels in dynamic graph learning through hyperbolic geometry. Its strong performance makes it a valuable addition to our approach.

Given feature vectors $X_t^E$ of snapshot $t$ in Euclidean space, an HGNN layer first adopts an exponential map to project Euclidean space vectors to hyperbolic space as follows $X_t^{\mathcal{H}} = exp^c(X_t^E)$, and then performs aggregation and activation similar to GNN but in a hyperbolic manner, $\tilde{X}_t^{\mathcal{H}} = \textbf{HGNN}(X_t^{\mathcal{H}})$. To prevent recurrent neural networks from only emphasizing the most nearby time and to ensure stability along with generalization of the embedding, HTGN uses temporal contextual attention (HTA) to generalize the lastest $w$ hidden states such that $\tilde{H}_{t-1}^{\mathcal{H}} = \textbf{HTA}(H_{t-w}; ...; H_{t-1})$ [62]. HGRU takes the outputs from HGNN, $\tilde{X}_t^{\mathcal{H}}$, and the attentive hidden state, $\tilde{H}_{t-1}^{\mathcal{H}}$, from HTA as input to update gates and memory cells and then provides the latest hidden state as the output, $H_t^{\mathcal{H}} = \textbf{HGRU}(\tilde{X}_t^{\mathcal{H}}, \tilde{H}_{t-1}^{\mathcal{H}})$. To interpret hyperbolic embeddings, [62] adopt Poincaré ball model with negative curve $-c$, given $c > 0$, coresponds to the Riemannian manifold $(\mathbb{H}^{n,c}) = \{x \in \mathbb{R}^n : c||x||^2 < 1\}$ is an open n-dimensional ball. Given a Euclidean space vector $x_i^E \in \mathbb{R}^d$, we consider it as a point in the tangent space $\mathcal{T}_{x'}\mathbb{H}^{d,c}$ and adopt the exponential map to project it into hyperbolic space :

$$x_i^{\mathcal{H}} = exp_{x'}^c(x_i^E) \tag{5}$$

Resulting in $x_i^{\mathcal{H}} \in \mathbb{H}^{d,c}$, which is then served as input to the HGNN layer as follows [62]:

$$\mathbf{m}_i^{\mathcal{H}} = W \otimes^c \mathbf{x}_i^{\mathcal{H}} \oplus^c \mathbf{b}, \tag{6a}$$

$$\tilde{\mathbf{m}}_i^{\mathcal{H}} = \exp_{\mathbf{x}'}^c \Big( \sum_{j \in \mathcal{N}(i)} \alpha_{ij} \log_{\mathbf{x}'}^c(\mathbf{m}_i^{\mathcal{H}}) \Big), \tag{6b}$$

$$\tilde{\mathbf{x}}_i^{\mathcal{H}} = \exp_{\mathbf{x}'}^c (\sigma(\log_{\mathbf{x}'}^c(\tilde{\mathbf{m}}_i^{\mathcal{H}})). \tag{6c}$$

where $W$, $b$ are learnable parameters and hyperbolic activation function $\sigma$ achieved by applying logarithmic and exponential mapping. HGNN leverages attention-based aggregation by assigning attention score $\alpha_{ij}$ to indicate the importance of neighbour $j$ to node $i$, computed as followed:

$$\alpha_{ij} = softmax_{(j \in \mathcal{N}(i))}(s_{ij}) = \frac{\exp(s_{ij})}{\sum_{j' \in \mathcal{N}_i} \exp(s_{ij'})}, \tag{7}$$
$$s_{ij} = \text{LeakReLU}(a^T[\log_0^c(m_i^l) \| \log_0^c(m_j^l)]),$$

where $a$ is trainable vector and $\|$ denotes concatenation operation.

The output of HGNN, $\tilde{X}_t^{\mathcal{H}}$, is then used as input to HGRU along with attentive hidden state $\tilde{H}_{t-1}^{\mathcal{H}}$ obtained by HTA, which generalize $H_{t-1}$ to lastest $w$ snapshots $\{H_{t-w}, ..., H_{t-1}\}$ [62]. Operations behind HGRU are characterized by the following equation [62]:

$$X_t^E = \log_{\mathbf{x}'}^c(\tilde{X}_t^{\mathcal{H}}), \tag{8a}$$

$$H_{t-1}^E = \log_{\mathbf{x}'}^c(\tilde{H}_{t-1}^{\mathcal{H}}), \tag{8b}$$

$$P_t^E = \sigma(W_z X_t^E + U_z H_{t-1}^E) \tag{8c}$$

$$R_t^E = \sigma(W_r X_t^E + U_r H_{t-1}^E), \tag{8d}$$

$$\tilde{H}_t^E = \tanh(W_h X_t^E + U_h(R_t \odot H_{t-1}^E)), \tag{8e}$$

$$H_t^E = (1 - P_t^E) \odot \tilde{H}_t^E + P_t^E \odot H_{t-1}^E, \tag{8f}$$

$$H_t^{\mathcal{H}} = \exp_{\mathbf{x}'}^c(H_t^E). \tag{8g}$$

where $W_z, W_r, W_h, U_z, U_r, U_h$ are the trainable weight matrices, $P_t^E$ is the update gate to control the output and $R_t^E$ is the reset gate to balance the input and memory [62].

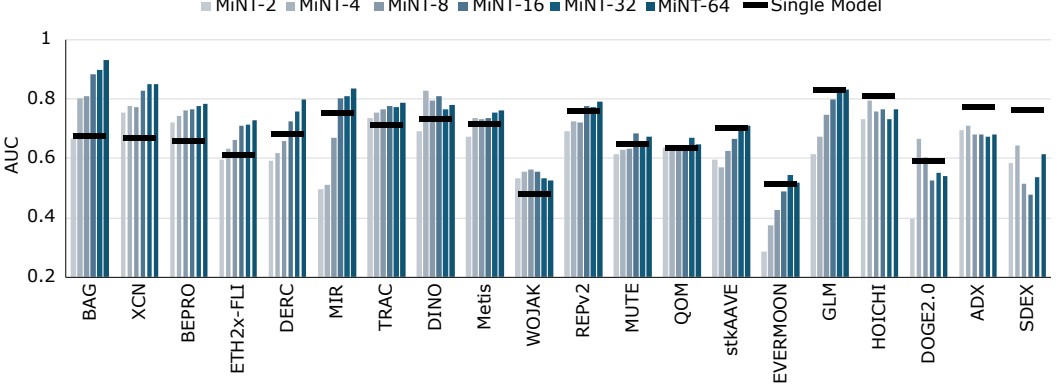

Figure 8: Test AUC-ROC of multi-network models trained on $2^n$ datasets where $n \in [1, 6]$ and evaluated on unseen test datasets for network growth or shrink task. Comparing the performance of single models trained and tested on each dataset.

# H    Social Domain Results

To assess the generalizability of our proposed MiNT models beyond transaction-based networks, we conducted experiments on a diverse set of eight real-world social interaction networks. This evaluation aims to demonstrate that MiNT is not constrained to transactional graph domains and can effectively transfer learned representations to structurally and semantically distinct networks.

The selected social datasets include *LastFM*[48], *MathOverflow*[39], *SuperUser*[39], *Email-Eu*[39], *AskUbuntu*[39], *CollegeMsg*[38], *StackOverflow*[39], and *RedditB*[26]. These datasets span a wide range of social communication settings, from question-answering platforms to messaging and collaboration networks, providing a rigorous testbed for cross-domain transfer.

Table 9: **AUC** scores of social multi-network models and single models on test sets across three seeds for the network growth or shrink task. Best scores per dataset are shown in **bold**.

| Network | Standard Training | Transfer Model | | |
|---|---|---|---|---|
| | HTGN | MiNT-2 Social | MiNT-4 Social | MiNT-6 Social |
| mathoverflow | **0.788 ± 0.050** | 0.750 ± 0.000 | 0.751 ± 0.000 | 0.758 ± 0.015 |
| RedditB | 0.655 ± 0.040 | 0.650 ± 0.002 | 0.651 ± 0.004 | **0.663 ± 0.008** |

We trained three variants of the MiNT model, MiNT-2, MiNT-4 and MiNT-6 on six social networks and evaluated them on two held-out unseen networks: *MathOverflow* and *RedditB*. As shown in Table 9, the transferable MiNT models perform competitively with the standard HTGN model that is trained directly on the test network. Notably, MiNT-6 achieves the best performance on *RedditB* (0.663 AUC), surpassing the standard HTGN model, and demonstrates strong results on *MathOverflow* (0.758 AUC), further closing the gap with the single model baseline. We observe a consistent scaling behavior with increasing model capacity (i.e., number of source networks), similar to what was reported in transaction network experiments. This trend indicates that as the number of training networks increases, the MiNT models are better equipped to capture structural and temporal patterns in unseen networks. This reinforces the model's ability to extract transferable knowledge and leverage broader training contexts effectively.

# I   MiNT on Additional Property Prediction Task

Table 10: **AUC** scores of multi-network models and single models on test sets across three seeds on the largest connected component growth task. Best results in **bold**, second best underlined.

| Network | Standard Training | Transfer Model | | | | |
|---|---|---|---|---|---|---|
| | HTGN | MiNT-4 | MiNT-8 | MiNT-16 | MiNT-32 | MiNT-64 |
| MIR | 0.745 ± 0.023 | 0.570 ± 0.117 | 0.655 ± 0.012 | 0.783 ± 0.041 | 0.766 ± 0.053 | **0.845 ± 0.035** |
| DOGE2.0 | 0.446 ± 0.164 | 0.530 ± 0.113 | 0.548 ± 0.063 | 0.631 ± 0.027 | 0.571 ± 0.139 | **0.661 ± 0.047** |
| MUTE | 0.574 ± 0.022 | 0.471 ± 0.014 | 0.468 ± 0.021 | 0.509 ± 0.037 | **0.592 ± 0.038** | 0.582 ± 0.078 |
| EVERMOON | 0.494 ± 0.127 | 0.424 ± 0.059 | 0.421 ± 0.029 | 0.376 ± 0.014 | **0.542 ± 0.077** | 0.527 ± 0.118 |
| DERC | 0.717 ± 0.035 | 0.552 ± 0.015 | 0.584 ± 0.040 | 0.554 ± 0.011 | **0.733 ± 0.067** | 0.689 ± 0.096 |
| ADX | **0.753 ± 0.013** | 0.610 ± 0.019 | 0.635 ± 0.033 | 0.603 ± 0.019 | 0.619 ± 0.012 | 0.587 ± 0.014 |
| HOICHI | **0.746 ± 0.010** | 0.738 ± 0.009 | 0.696 ± 0.072 | 0.715 ± 0.027 | 0.592 ± 0.147 | 0.722 ± 0.034 |
| SDEX | **0.911 ± 0.104** | 0.330 ± 0.117 | 0.425 ± 0.199 | 0.361 ± 0.113 | 0.437 ± 0.316 | 0.382 ± 0.280 |
| BAG | 0.493 ± 0.043 | 0.772 ± 0.213 | 0.685 ± 0.163 | 0.892 ± 0.036 | **0.952 ± 0.019** | 0.893 ± 0.074 |
| XCN | 0.566 ± 0.199 | 0.742 ± 0.039 | 0.688 ± 0.041 | 0.802 ± 0.037 | 0.774 ± 0.144 | **0.827 ± 0.025** |
| ETH2x-FLI | 0.561 ± 0.037 | 0.610 ± 0.015 | 0.625 ± 0.020 | **0.658 ± 0.018** | 0.636 ± 0.076 | 0.618 ± 0.025 |
| stkAAVE | 0.623 ± 0.077 | 0.613 ± 0.041 | 0.567 ± 0.038 | 0.668 ± 0.061 | 0.687 ± 0.045 | **0.688 ± 0.019** |
| GLM | 0.761 ± 0.031 | 0.585 ± 0.144 | 0.679 ± 0.026 | 0.698 ± 0.054 | 0.783 ± 0.031 | **0.818 ± 0.074** |
| QOM | 0.658 ± 0.150 | 0.535 ± 0.036 | 0.513 ± 0.003 | 0.566 ± 0.036 | **0.696 ± 0.092** | 0.645 ± 0.109 |
| WOJAK | 0.378 ± 0.028 | 0.407 ± 0.012 | 0.362 ± 0.053 | 0.384 ± 0.024 | 0.421 ± 0.029 | **0.492 ± 0.107** |
| DINO | 0.706 ± 0.120 | 0.794 ± 0.090 | **0.827 ± 0.039** | 0.815 ± 0.043 | 0.753 ± 0.165 | 0.561 ± 0.006 |
| Metis | 0.679 ± 0.039 | 0.697 ± 0.031 | 0.671 ± 0.047 | 0.711 ± 0.028 | 0.705 ± 0.047 | **0.780 ± 0.041** |
| REPv2 | 0.730 ± 0.007 | 0.653 ± 0.014 | 0.642 ± 0.061 | 0.694 ± 0.002 | **0.765 ± 0.030** | 0.742 ± 0.041 |
| TRAC | 0.733 ± 0.009 | 0.658 ± 0.040 | 0.643 ± 0.052 | 0.720 ± 0.048 | **0.767 ± 0.012** | 0.762 ± 0.028 |
| BEPRO | **0.694 ± 0.009** | 0.587 ± 0.002 | 0.604 ± 0.006 | 0.589 ± 0.018 | 0.601 ± 0.129 | 0.628 ± 0.017 |
| **Top Rank ↑** | 4 | 0 | 1 | 1 | 6 | **7** |
| Avg. Rank ↓ | 2.80 | 4.40 | 4.65 | 3.45 | 2.50 | **2.20** |

To further demonstrate that the scalability of our approach is not restricted to a specific property, we extended our experiments to evaluate the performance of MiNT models on a new task. This task involves predicting the growth or shrinkage of the largest connected component, which is particularly meaningful in the context of transaction networks.

**Experimental Results.** Table 10 presents the performance of MiNT models and the baseline HTGN across twenty networks. MiNT models, especially MiNT-32 and MiNT-64, outperform the baseline in the majority of cases. MiNT 64 achieves the highest AUC in seven networks and ranks second in three others. It also records the best average rank overall, indicating strong generalization to this new property prediction task.

These results show that MiNT models are not limited to a particular type of graph signal. Instead, they are capable of adapting to a broad range of temporal properties.

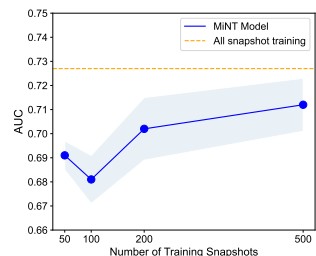

## J Effect of Snapshot Scaling on Model Performance

We conducted an additional experiment to analyze how the number of training snapshots affects model performance over time. Specifically, we evaluated the scaling behavior of the MiNT-64 model by training it on different amounts of historical data. For this study, we selected five snapshot counts: 50, 100, 200, 500, and full snapshot history. These snapshots were drawn sequentially from the end of each dataset, just before the validation period, to simulate a realistic training scenario.

Figure 9: Scaling effect of number of snapshots used in MiNT-64 training.

For each configuration, MiNT-64 was trained using three random seeds, and the average AUC results are presented in Figure 9. The trend illustrates the scaling behavior of the model as more snapshots are provided. As the training window expands, the model gains access to richer temporal information, which contributes to improved generalization and performance. The trend suggests that access to a larger number of historical snapshots enables temporal models to better capture evolving patterns and improve predictive performance.

## K Effect Of Data Selection

We investigate the effect of data selection on the performance of MiNT models trained with different training data packs. As the first work on multi-network training for temporal graphs, we explore the importance of our dataset selection process. To avoid any bias, we randomly sampled the training datasets from the 64 available networks. We conducted an empirical experiment to examine the impact of dataset selection on training MiNT models. In this experiment, we choose three disjoint sets of datasets (data pack A, B, and C) for training MiNT-2, MiNT-4, MiNT-8, and MiNT-16 and two disjoint sets of datasets (data pack A, B) for training MiNT-32. Using disjoint data packs ensures that each model is trained on unique data, eliminating any overlap that could obscure the results. We then test our models on 20 unseen test datasets. Note that MiNT-32 has only two packs, whereas other MiNT

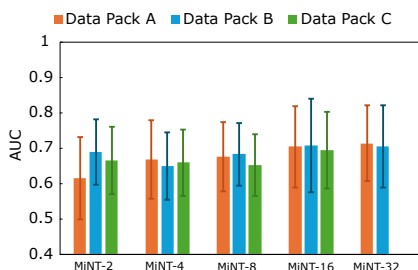

Figure 10: Effect of data selection on model performance for network growth or shrink task. When models reach larger sizes (i.e., Mint-32), the effect of data packs is negligibly similar.

models have three packs due to the limited number of available training networks. Specifically, since MiNT-32 requires 32 distinct networks per training run and only 64 total networks are available, we can only create two non-overlapping training sets of size 32. In contrast, smaller models such as MiNT-2 through MiNT-16 allow for more disjoint groupings.

As shown in Figure 10, as the number of training networks increases, the multi-network model performance increases while the variance between different choices of training networks decreases. However, the difference between models that use the same number of datasets diminishes as we move from models of 2 to 32 datasets. We observe that smaller models (i.e., MiNT-2) have a higher variance when compared to larger models (i.e., MiNT-64); in addition, the model performance also increases from small to large models. For example, MiNT-64 outperforms MiNT-32 on 16 out of 20 datasets.

## L Choice of graph pooling

Pooling plays an important role in temporal graph property prediction. In our study, we employ mean pooling due to its stability across diverse datasets. To assess alternative choices, we compared mean pooling with max pooling and found that the latter leads to an average 6% drop in AUC, Table 11, highlighting the effectiveness of mean pooling in our zero-shot temporal graph setting. Adaptive or hierarchical pooling mechanisms may better capture structural dependencies, and we consider this an interesting direction for future work.

## M  Additional Results

Here, we present the test results for the six multi-network models trained on different network sizes, as well as the single model results. Figure 8 illustrates the AUC of these models on the test set. In most datasets, multi-network models outperform the single model. We have also compared our model against additional state-of-the-art models, specifically including Roland [64], EvolveGCN [40], GC-LSTM [11], and the only model designed for temporal graph properties prediction, GraphPulse [50] as baselines for the test set. In Table 12 and Table 13 the average and standard deviation of AUC and AP are presented, respectively, for all models. Surprisingly, MiNT-64 stands out as the best model, consistently achieving competitive performance in a greater number of datasets for both AUC and AP scores compared to all other models. Similarly, MN-32 demonstrates strong performance, achieving the highest score in several datasets and placing second in numerous others; however, it does not surpass MN-64 in overall rankings. These results show the power of multi-network models in performing downstream tasks on

Table 11: Test AUC on unseen networks between MiNT-4 with max pooling and MiNT-4 with mean pooling

| Dataset | MiNT-4 Test AUC Max Pooling | MiNT-4 Test AUC Mean Pooling |
|---|---|---|
| MIR | **0.588** | 0.510 |
| DOGE2.0 | 0.500 | **0.667** |
| MUTE | **0.685** | 0.627 |
| EVERMOON | **0.622** | 0.373 |
| DERC | **0.761** | 0.617 |
| ADX | 0.692 | **0.708** |
| HOICHI | 0.583 | **0.795** |
| SDEX | 0.401 | **0.643** |
| BAG | 0.610 | **0.802** |
| XCN | 0.557 | **0.774** |
| ETH2x-FLI | **0.704** | 0.632 |
| stkAAVE | 0.525 | **0.571** |
| GLM | 0.598 | **0.671** |
| QOM | 0.611 | **0.624** |
| WOJAK | **0.611** | 0.556 |
| DINO | 0.480 | **0.827** |
| Metis | 0.558 | **0.734** |
| REPv2 | **0.746** | 0.725 |
| TRAC | 0.572 | **0.752** |
| BEPRO | **0.788** | 0.742 |
| **Average** | 0.6096 | **0.668** |

unseen datasets. Importantly, this high level of performance is achieved through zero-shot inference, meaning that the model was not specifically trained on these datasets. In contrast, other models, including GraphPulse, were trained directly for the datasets they evaluated. This considerable difference underscores the potential of MiNT-64 and highlights the power of zero-shot learning in effectively leveraging knowledge across different temporal graphs.

Table 14 presents the detailed performance of all MiNT models trained with GCLSTM. Notably, we observed a consistent trend with GCLSTM: as the model was trained on a larger number of networks, its zero-shot inference performance improved significantly. This highlights the positive impact of training on diverse networks for enhancing the model's generalization capabilities.

Table 12: **AUC** scores of multi-network models and single models on test sets across three seeds, including comparisons with state-of-the-art models EvolveGCN, GC-LSTM, GraphPulse and ROLAND for network growth or shrink task. The best performance is shown in bold, and the second best is underlined.

| Token | ROLAND | GraphPulse | HTGN | GCLSTM | EvolveGCN | MiNT-2 | MiNT-4 | MiNT-8 | MiNT-16 | MiNT-32 | MiNT-64 |
|---|---|---|---|---|---|---|---|---|---|---|---|
| WOJAK | 0.529 ± 0.005 | 0.467 ± 0.030 | 0.479 ± 0.005 | 0.484 ± 0.000 | 0.505 ± 0.023 | 0.534 ± 0.020 | 0.556 ± 0.029 | **0.561 ± 0.018** | 0.556 ± 0.016 | 0.534 ± 0.017 | 0.524 ± 0.027 |
| DOGE2.0 | 0.513 ± 0.022 | 0.384 ± 0.180 | 0.590 ± 0.059 | 0.538 ± 0.000 | 0.551 ± 0.022 | 0.397 ± 0.124 | **0.667 ± 0.219** | 0.603 ± 0.080 | 0.526 ± 0.059 | 0.551 ± 0.022 | 0.538 ± 0.038 |
| EVERMOON | 0.349 ± 0.119 | 0.519 ± 0.130 | 0.512 ± 0.023 | **0.562 ± 0.179** | 0.451 ± 0.046 | 0.373 ± 0.037 | 0.426 ± 0.065 | 0.488 ± 0.054 | 0.543 ± 0.075 | 0.517 ± 0.039 | |
| QOM | 0.641 ± 0.003 | **0.775 ± 0.011** | 0.633 ± 0.017 | 0.612 ± 0.001 | 0.618 ± 0.002 | 0.635 ± 0.061 | 0.624 ± 0.025 | 0.633 ± 0.032 | 0.644 ± 0.009 | 0.669 ± 0.034 | 0.647 ± 0.019 |
| SDEX | 0.483 ± 0.254 | **0.762 ± 0.034** | 0.720 ± 0.002 | 0.733 ± 0.028 | 0.585 ± 0.119 | 0.643 ± 0.021 | 0.515 ± 0.031 | 0.476 ± 0.010 | 0.536 ± 0.042 | 0.614 ± 0.020 | |
| ETH2x-FLI | 0.621 ± 0.023 | 0.666 ± 0.047 | 0.610 ± 0.059 | 0.670 ± 0.009 | 0.688 ± 0.010 | 0.595 ± 0.083 | 0.632 ± 0.019 | 0.663 ± 0.018 | 0.710 ± 0.037 | 0.715 ± 0.032 | **0.729 ± 0.015** |
| BEPRO | 0.439 ± 0.125 | **0.783 ± 0.003** | 0.655 ± 0.038 | 0.632 ± 0.019 | 0.610 ± 0.012 | 0.720 ± 0.028 | 0.742 ± 0.013 | 0.762 ± 0.007 | 0.765 ± 0.024 | 0.776 ± 0.008 | 0.782 ± 0.003 |
| XCN | 0.765 ± 0.015 | 0.821 ± 0.004 | 0.668 ± 0.099 | 0.306 ± 0.092 | 0.512 ± 0.067 | 0.754 ± 0.025 | 0.774 ± 0.062 | 0.773 ± 0.076 | 0.827 ± 0.061 | 0.848 ± 0.000 | **0.851 ± 0.043** |
| BAG | 0.418 ± 0.016 | **0.934 ± 0.020** | 0.673 ± 0.227 | 0.196 ± 0.179 | 0.329 ± 0.040 | 0.667 ± 0.134 | 0.802 ± 0.155 | 0.808 ± 0.095 | 0.884 ± 0.044 | 0.898 ± 0.075 | 0.931 ± 0.028 |
| TRAC | 0.495 ± 0.223 | 0.767 ± 0.001 | 0.712 ± 0.071 | 0.748 ± 0.000 | 0.748 ± 0.000 | 0.734 ± 0.012 | 0.752 ± 0.009 | 0.764 ± 0.012 | 0.776 ± 0.012 | 0.770 ± 0.007 | **0.785 ± 0.008** |
| DERC | 0.405 ± 0.357 | 0.769 ± 0.045 | 0.683 ± 0.013 | 0.703 ± 0.022 | 0.669 ± 0.009 | 0.593 ± 0.104 | 0.617 ± 0.030 | 0.657 ± 0.009 | 0.723 ± 0.058 | 0.756 ± 0.045 | **0.798 ± 0.027** |
| Metis | 0.696 ± 0.108 | **0.812 ± 0.011** | 0.715 ± 0.122 | 0.646 ± 0.023 | 0.688 ± 0.027 | 0.672 ± 0.103 | 0.734 ± 0.017 | 0.730 ± 0.036 | 0.734 ± 0.016 | 0.753 ± 0.005 | 0.760 ± 0.025 |
| REPv2 | 0.751 ± 0.003 | **0.830 ± 0.001** | 0.760 ± 0.012 | 0.725 ± 0.014 | 0.709 ± 0.002 | 0.690 ± 0.024 | 0.725 ± 0.023 | 0.719 ± 0.022 | 0.774 ± 0.013 | 0.773 ± 0.013 | 0.789 ± 0.020 |
| DINO | 0.497 ± 0.092 | 0.801 ± 0.004 | 0.730 ± 0.195 | **0.874 ± 0.028** | 0.868 ± 0.029 | 0.692 ± 0.140 | 0.827 ± 0.112 | 0.794 ± 0.096 | 0.809 ± 0.087 | 0.764 ± 0.048 | 0.779 ± 0.113 |
| HOICHI | **0.815 ± 0.036** | 0.714 ± 0.010 | 0.807 ± 0.047 | 0.857 ± 0.000 | 0.856 ± 0.001 | 0.733 ± 0.101 | 0.795 ± 0.025 | 0.759 ± 0.040 | 0.763 ± 0.026 | 0.731 ± 0.029 | 0.765 ± 0.018 |
| MUTE | 0.289 ± 0.042 | **0.779 ± 0.004** | 0.649 ± 0.015 | 0.593 ± 0.030 | 0.617 ± 0.010 | 0.613 ± 0.027 | 0.627 ± 0.024 | 0.633 ± 0.024 | 0.684 ± 0.042 | 0.657 ± 0.035 | 0.673 ± 0.013 |
| GLM | 0.559 ± 0.357 | 0.769 ± 0.018 | 0.830 ± 0.029 | 0.451 ± 0.003 | 0.501 ± 0.033 | 0.613 ± 0.115 | 0.671 ± 0.034 | 0.746 ± 0.082 | 0.800 ± 0.062 | 0.826 ± 0.035 | **0.831 ± 0.024** |
| MIR | 0.228 ± 0.060 | 0.689 ± 0.097 | 0.750 ± 0.005 | 0.768 ± 0.026 | 0.745 ± 0.015 | 0.497 ± 0.192 | 0.510 ± 0.015 | 0.669 ± 0.103 | 0.800 ± 0.044 | 0.809 ± 0.022 | **0.836 ± 0.016** |
| stkAAVE | 0.591 ± 0.122 | **0.743 ± 0.006** | 0.702 ± 0.042 | 0.368 ± 0.011 | 0.397 ± 0.022 | 0.597 ± 0.076 | 0.571 ± 0.026 | 0.626 ± 0.023 | 0.666 ± 0.033 | 0.696 ± 0.027 | 0.709 ± 0.022 |
| ADX | **0.761 ± 0.011** | 0.784 ± 0.002 | 0.769 ± 0.018 | 0.723 ± 0.002 | 0.718 ± 0.004 | 0.695 ± 0.003 | 0.708 ± 0.025 | 0.680 ± 0.008 | 0.678 ± 0.019 | 0.671 ± 0.015 | 0.679 ± 0.024 |

## N  Zero-Shot Inference Efficiency of MiNT

As shown in Table 15, MiNT demonstrates remarkable computational efficiency across all unseen datasets compared to training a single HTGN model on each individual unseen network. On average, HTGN requires 2141.66 seconds to train a model per dataset, whereas MiNT completes inference in

Table 13: **AP** scores of multi-network models, and single models on test sets across three seeds, including comparisons with state-of-the-art models EvolveGCN, GC-LSTM, GraphPulse, and Roland for the network growth or shrink task. The best performance is shown in bold, and the second best is underlined.

| Token | ROLAND | GraphPulse | HTGN | GCLSTM | EvolveGCN | MiNT-2 | MiNT-4 | MiNT-8 | MiNT-16 | MiNT-32 | MiNT-64 |
|---|---|---|---|---|---|---|---|---|---|---|---|
| WOJAK | 0.844 ± 0.003 | **0.863** ± 0.006 | 0.812 ± 0.003 | 0.812 ± 0.000 | 0.827 ± 0.017 | 0.832 ± 0.009 | 0.836 ± 0.015 | 0.842 ± 0.015 | 0.850 ± 0.006 | 0.842 ± 0.008 | 0.837 ± 0.019 |
| DOGE2.0 | 0.918 ± 0.006 | **0.966** ± 0.002 | 0.933 ± 0.010 | 0.925 ± 0.000 | 0.927 ± 0.004 | 0.889 ± 0.031 | 0.940 ± 0.050 | 0.936 ± 0.014 | 0.920 ± 0.014 | 0.927 ± 0.004 | 0.921 ± 0.014 |
| EVERMOON | 0.390 ± 0.033 | **0.768** ± 0.01 | 0.585 ± 0.065 | 0.612 ± 0.200 | 0.494 ± 0.017 | 0.442 ± 0.059 | 0.508 ± 0.045 | 0.542 ± 0.031 | 0.530 ± 0.040 | 0.567 ± 0.053 | 0.551 ± 0.021 |
| QOM | 0.624 ± 0.004 | 0.840 ± 0.002 | 0.623 ± 0.024 | 0.592 ± 0.001 | 0.597 ± 0.002 | 0.632 ± 0.070 | 0.617 ± 0.022 | 0.616 ± 0.007 | 0.626 ± 0.020 | **0.648** ± 0.027 | 0.635 ± 0.027 |
| SDEX | 0.631 ± 0.133 | 0.662 ± 0.017 | **0.825** ± 0.048 | 0.725 ± 0.002 | 0.750 ± 0.025 | 0.723 ± 0.019 | 0.725 ± 0.021 | 0.650 ± 0.046 | 0.628 ± 0.036 | 0.697 ± 0.064 | 0.699 ± 0.021 |
| ETH2x-FLI | 0.619 ± 0.077 | **0.836** ± 0.015 | 0.590 ± 0.103 | 0.735 ± 0.018 | 0.756 ± 0.013 | 0.607 ± 0.122 | 0.621 ± 0.039 | 0.658 ± 0.057 | 0.745 ± 0.051 | 0.737 ± 0.049 | 0.784 ± 0.007 |
| BEPRO | 0.513 ± 0.080 | 0.802 ± 0.001 | 0.686 ± 0.042 | 0.637 ± 0.022 | 0.622 ± 0.009 | 0.769 ± 0.015 | 0.799 ± 0.016 | 0.804 ± 0.034 | 0.815 ± 0.007 | **0.816** ± 0.014 | |
| XCN | 0.747 ± 0.037 | 0.793 ± 0.002 | 0.687 ± 0.085 | 0.420 ± 0.032 | 0.555 ± 0.073 | 0.708 ± 0.065 | 0.765 ± 0.080 | 0.781 ± 0.082 | 0.829 ± 0.057 | 0.851 ± 0.023 | **0.861** ± 0.042 |
| BAG | 0.289 ± 0.005 | **0.957** ± 0.004 | 0.523 ± 0.290 | 0.235 ± 0.041 | 0.263 ± 0.011 | 0.474 ± 0.152 | 0.699 ± 0.193 | 0.682 ± 0.160 | 0.784 ± 0.118 | 0.829 ± 0.119 | 0.889 ± 0.043 |
| TRAC | 0.499 ± 0.192 | **0.767** ± 0.002 | 0.685 ± 0.074 | 0.716 ± 0.006 | 0.722 ± 0.001 | 0.705 ± 0.013 | 0.734 ± 0.012 | 0.741 ± 0.006 | 0.764 ± 0.015 | 0.741 ± 0.015 | 0.758 ± 0.021 |
| DERC | 0.460 ± 0.296 | **0.773** ± 0.004 | 0.532 ± 0.021 | 0.621 ± 0.053 | 0.513 ± 0.012 | 0.505 ± 0.157 | 0.477 ± 0.021 | 0.516 ± 0.030 | 0.639 ± 0.118 | 0.700 ± 0.080 | 0.741 ± 0.024 |
| Metis | 0.596 ± 0.120 | **0.801** ± 0.003 | 0.601 ± 0.187 | 0.575 ± 0.041 | 0.577 ± 0.006 | 0.532 ± 0.126 | 0.645 ± 0.029 | 0.632 ± 0.056 | 0.611 ± 0.021 | 0.647 ± 0.026 | 0.639 ± 0.077 |
| REPv2 | 0.727 ± 0.003 | **0.797** ± 0.003 | 0.758 ± 0.033 | 0.691 ± 0.006 | 0.689 ± 0.001 | 0.610 ± 0.063 | 0.619 ± 0.019 | 0.635 ± 0.042 | 0.705 ± 0.027 | 0.721 ± 0.004 | 0.729 ± 0.011 |
| DINO | 0.591 ± 0.076 | 0.871 ± 0.026 | 0.747 ± 0.175 | **0.881** ± 0.029 | 0.875 ± 0.024 | 0.738 ± 0.113 | 0.842 ± 0.102 | 0.793 ± 0.094 | 0.824 ± 0.077 | 0.753 ± 0.030 | 0.765 ± 0.119 |
| HOICHI | **0.699** ± 0.031 | 0.623 ± 0.003 | 0.666 ± 0.062 | 0.650 ± 0.000 | 0.658 ± 0.011 | 0.531 ± 0.109 | 0.677 ± 0.049 | 0.605 ± 0.037 | 0.609 ± 0.016 | 0.551 ± 0.045 | 0.594 ± 0.012 |
| MUTE | 0.332 ± 0.012 | **0.726** ± 0.002 | 0.615 ± 0.049 | 0.504 ± 0.012 | 0.527 ± 0.015 | 0.579 ± 0.022 | 0.612 ± 0.041 | 0.603 ± 0.058 | 0.675 ± 0.032 | 0.609 ± 0.021 | 0.647 ± 0.048 |
| GLM | 0.585 ± 0.191 | 0.712 ± 0.047 | 0.797 ± 0.024 | 0.513 ± 0.001 | 0.529 ± 0.013 | 0.598 ± 0.123 | 0.651 ± 0.031 | 0.709 ± 0.088 | 0.783 ± 0.092 | 0.819 ± 0.035 | **0.838** ± 0.032 |
| MIR | 0.317 ± 0.019 | 0.766 ± 0.041 | 0.751 ± 0.003 | 0.765 ± 0.012 | 0.752 ± 0.007 | 0.493 ± 0.212 | 0.442 ± 0.024 | 0.645 ± 0.133 | 0.783 ± 0.064 | 0.799 ± 0.015 | **0.811** ± 0.019 |
| stkAAVE | 0.630 ± 0.109 | **0.751** ± 0.005 | 0.750 ± 0.020 | 0.506 ± 0.003 | 0.493 ± 0.009 | 0.662 ± 0.066 | 0.622 ± 0.011 | 0.694 ± 0.021 | 0.730 ± 0.037 | 0.741 ± 0.020 | 0.759 ± 0.019 |
| ADX | 0.738 ± 0.026 | **0.765** ± 0.003 | 0.758 ± 0.017 | 0.666 ± 0.002 | 0.661 ± 0.017 | 0.638 ± 0.021 | 0.667 ± 0.040 | 0.632 ± 0.010 | 0.621 ± 0.013 | 0.622 ± 0.018 | 0.628 ± 0.012 |

Table 14: **AP and AUC** scores of GCLSTM-based multi-network models on test sets across three seeds for network growth or shrink task. The best performance is shown in bold, and the second best is underlined.

| Token | AUC | | | | | | AP | | | | | |
|---|---|---|---|---|---|---|---|---|---|---|---|---|
| | MiNT-2 | MiNT-4 | MiNT-8 | MiNT-16 | MiNT-32 | MiNT-64 | MiNT-2 | MiNT-4 | MiNT-8 | MiNT-16 | MiNT-32 | MiNT-64 |
| MIR | 0.653 ± 0.154 | 0.638 ± 0.090 | 0.588 ± 0.135 | 0.765 ± 0.049 | 0.742 ± 0.036 | **0.789** ± 0.016 | 0.667 ± 0.153 | 0.602 ± 0.134 | 0.550 ± 0.166 | 0.750 ± 0.019 | 0.758 ± 0.016 | **0.777** ± 0.013 |
| DOGE2 | 0.487 ± 0.089 | 0.590 ± 0.146 | 0.487 ± 0.219 | 0.282 ± 0.097 | **0.769** ± 0.133 | 0.551 ± 0.022 | 0.910 ± 0.019 | 0.930 ± 0.030 | 0.907 ± 0.046 | 0.839 ± 0.057 | **0.965** ± 0.024 | 0.927 ± 0.004 |
| MUTE | 0.592 ± 0.076 | 0.627 ± 0.018 | 0.561 ± 0.035 | 0.589 ± 0.009 | 0.627 ± 0.009 | **0.636** ± 0.003 | 0.534 ± 0.056 | 0.555 ± 0.017 | 0.502 ± 0.022 | 0.501 ± 0.006 | 0.563 ± 0.002 | **0.568** ± 0.002 |
| EVERMOON | 0.429 ± 0.078 | 0.318 ± 0.152 | 0.306 ± 0.085 | 0.315 ± 0.154 | 0.420 ± 0.084 | **0.494** ± 0.048 | 0.493 ± 0.095 | 0.423 ± 0.097 | 0.427 ± 0.037 | 0.447 ± 0.123 | 0.530 ± 0.048 | **0.560** ± 0.010 |
| DERC | 0.614 ± 0.129 | 0.618 ± 0.085 | 0.569 ± 0.085 | **0.736** ± 0.027 | 0.647 ± 0.054 | 0.696 ± 0.011 | 0.647 ± 0.026 | 0.546 ± 0.113 | 0.460 ± 0.078 | **0.693** ± 0.032 | 0.559 ± 0.087 | 0.629 ± 0.012 |
| ADX | **0.692** ± 0.007 | 0.605 ± 0.182 | 0.674 ± 0.008 | 0.676 ± 0.003 | 0.678 ± 0.004 | 0.674 ± 0.022 | 0.614 ± 0.011 | 0.583 ± 0.147 | **0.634** ± 0.024 | 0.609 ± 0.005 | 0.617 ± 0.004 | 0.611 ± 0.010 |
| HOICHI | 0.663 ± 0.312 | 0.793 ± 0.065 | 0.633 ± 0.197 | 0.817 ± 0.010 | 0.816 ± 0.043 | **0.847** ± 0.046 | 0.541 ± 0.150 | 0.602 ± 0.066 | 0.471 ± 0.178 | 0.637 ± 0.016 | 0.630 ± 0.055 | **0.656** ± 0.014 |
| SDEX | 0.619 ± 0.210 | 0.721 ± 0.032 | 0.574 ± 0.233 | **0.741** ± 0.014 | 0.717 ± 0.020 | 0.724 ± 0.002 | 0.678 ± 0.115 | 0.732 ± 0.024 | 0.670 ± 0.092 | **0.752** ± 0.007 | 0.728 ± 0.009 | 0.729 ± 0.002 |
| BAG | **0.573** ± 0.072 | 0.525 ± 0.010 | 0.374 ± 0.029 | 0.442 ± 0.039 | 0.469 ± 0.060 | 0.529 ± 0.023 | **0.358** ± 0.036 | 0.334 ± 0.005 | 0.277 ± 0.010 | 0.303 ± 0.013 | 0.311 ± 0.025 | 0.337 ± 0.009 |
| XCN | **0.753** ± 0.026 | 0.739 ± 0.005 | 0.726 ± 0.014 | 0.736 ± 0.006 | 0.731 ± 0.005 | 0.733 ± 0.003 | **0.690** ± 0.064 | 0.657 ± 0.009 | 0.665 ± 0.031 | 0.656 ± 0.007 | 0.650 ± 0.003 | 0.653 ± 0.002 |
| ETH2x-FLI | 0.621 ± 0.119 | 0.615 ± 0.074 | 0.542 ± 0.086 | 0.675 ± 0.008 | 0.666 ± 0.021 | **0.697** ± 0.010 | 0.669 ± 0.084 | 0.570 ± 0.154 | 0.752 ± 0.015 | 0.669 ± 0.066 | 0.747 ± 0.021 | **0.766** ± 0.006 |
| stkAAVE | 0.601 ± 0.121 | 0.573 ± 0.084 | 0.517 ± 0.071 | 0.609 ± 0.032 | 0.624 ± 0.017 | **0.650** ± 0.028 | 0.687 ± 0.101 | 0.616 ± 0.108 | 0.571 ± 0.045 | 0.669 ± 0.066 | 0.710 ± 0.017 | **0.736** ± 0.022 |
| GLM | 0.448 ± 0.097 | 0.363 ± 0.112 | 0.331 ± 0.083 | **0.563** ± 0.016 | 0.463 ± 0.011 | 0.502 ± 0.027 | 0.467 ± 0.041 | 0.436 ± 0.012 | 0.437 ± 0.020 | **0.541** ± 0.010 | 0.480 ± 0.026 | 0.490 ± 0.012 |
| QOM | 0.594 ± 0.043 | 0.613 ± 0.009 | 0.574 ± 0.030 | 0.614 ± 0.005 | 0.614 ± 0.007 | **0.618** ± 0.004 | 0.587 ± 0.020 | 0.598 ± 0.004 | 0.573 ± 0.018 | 0.596 ± 0.005 | 0.597 ± 0.005 | **0.599** ± 0.003 |
| WOJAK | 0.516 ± 0.040 | 0.524 ± 0.016 | 0.561 ± 0.026 | 0.489 ± 0.060 | **0.598** ± 0.075 | 0.534 ± 0.020 | 0.810 ± 0.052 | 0.838 ± 0.014 | 0.834 ± 0.016 | 0.808 ± 0.027 | **0.862** ± 0.026 | 0.844 ± 0.007 |
| DINO | 0.667 ± 0.138 | 0.695 ± 0.147 | **0.738** ± 0.047 | 0.617 ± 0.148 | 0.704 ± 0.065 | 0.659 ± 0.039 | 0.719 ± 0.120 | 0.740 ± 0.107 | **0.746** ± 0.083 | 0.619 ± 0.073 | 0.683 ± 0.046 | 0.643 ± 0.041 |
| Metis | 0.692 ± 0.023 | 0.677 ± 0.030 | 0.609 ± 0.025 | 0.674 ± 0.020 | 0.690 ± 0.046 | **0.697** ± 0.043 | 0.558 ± 0.029 | 0.541 ± 0.043 | 0.485 ± 0.046 | 0.555 ± 0.019 | **0.586** ± 0.022 | 0.564 ± 0.019 |
| REPv2 | 0.670 ± 0.053 | 0.686 ± 0.043 | 0.706 ± 0.040 | **0.735** ± 0.017 | 0.707 ± 0.019 | 0.733 ± 0.019 | 0.617 ± 0.080 | 0.633 ± 0.008 | 0.709 ± 0.020 | **0.720** ± 0.031 | 0.654 ± 0.032 | 0.683 ± 0.022 |
| TRAC | 0.736 ± 0.015 | 0.736 ± 0.014 | 0.710 ± 0.027 | 0.741 ± 0.003 | **0.742** ± 0.004 | 0.741 ± 0.007 | 0.702 ± 0.031 | 0.709 ± 0.020 | 0.708 ± 0.016 | **0.720** ± 0.002 | 0.717 ± 0.003 | 0.717 ± 0.005 |
| BEPRO | 0.723 ± 0.053 | 0.720 ± 0.035 | 0.685 ± 0.069 | 0.734 ± 0.016 | **0.757** ± 0.015 | 0.746 ± 0.015 | 0.730 ± 0.096 | 0.755 ± 0.019 | 0.727 ± 0.063 | 0.764 ± 0.021 | **0.791** ± 0.007 | 0.776 ± 0.012 |

only 11.52 seconds, yielding an impressive average efficiency ratio of $180.86\times$. This result highlights the clear advantage of MiNT 's zero-shot inference capability. Once pretrained on multiple networks, it can generalize to unseen temporal graphs without the need for retraining, thus saving substantial computational resources.

A closer look at the dataset-level results reveals consistent and significant efficiency gains across all cases. Particularly, datasets such as DERC ($303.45\times$), DOGE2.0 ($266.78\times$), WOJAK ($249.11\times$), BEPRO ($240.73\times$), HOICHI ($220.80\times$), ADX ($221.69\times$), and TRAC ($220.39\times$) exhibit extremely large efficiency ratios, with MiNT inference being more than two hundred times faster than training a new HTGN model. These datasets tend to have relatively complex temporal dynamics or larger network sizes, implying that MiNT's pretraining enables it to generalize efficiently without the expensive retraining process required by HTGN.

Even for datasets where the efficiency ratio is relatively smaller, such as DINO ($91.87\times$), XCN ($90.65\times$), and GLM ($123.58\times$), the improvement still amounts to nearly two orders of magnitude,

Table 15: Comparison of time efficiency on unseen datasets (in seconds): HTGN vs. MiNT

| Dataset | HTGN Single Model Train Time | MiNT Inference Time | Efficiency Ratio |
|---|---|---|---|
| EVERMOON | 196.18 | 2.03 | 96.64 |
| DOGE2.0 | 392.16 | 1.47 | 266.78 |
| SDEX | 516.29 | 3.21 | 160.84 |
| BAG | 504.99 | 4.06 | 124.38 |
| DINO | 451.06 | 4.91 | 91.87 |
| WOJAK | 665.12 | 2.67 | 249.11 |
| XCN | 755.11 | 8.33 | 90.65 |
| HOICHI | 1262.99 | 5.72 | 220.80 |
| Metis | 1359.59 | 12.93 | 105.15 |
| QOM | 1681.95 | 8.34 | 201.67 |
| MUTE | 1679.20 | 13.74 | 122.21 |
| GLM | 1896.96 | 15.35 | 123.58 |
| ETH2x-FLI | 2931.61 | 13.47 | 217.64 |
| REPv2 | 3275.41 | 16.70 | 196.13 |
| DERC | 3486.62 | 11.49 | 303.45 |
| BEPRO | 3789.14 | 15.74 | 240.73 |
| stkAAVE | 3194.12 | 15.81 | 202.03 |
| ADX | 3673.39 | 16.57 | 221.69 |
| MIR | 4525.21 | 28.04 | 161.38 |
| TRAC | 6596.15 | 29.93 | 220.39 |
| **Average** | **2141.66** | **11.52** | **180.86** |

representing a dramatic reduction in computational cost. This consistency across diverse datasets underscores MiNT 's scalability and robustness.

Overall, these findings emphasize that MiNT not only provides dramatic time savings but also scales effectively across both small and large networks, maintaining reliable inference speed without sacrificing model performance. The ability to perform inference hundreds of times faster makes MiNT particularly advantageous in dynamic, real-world scenarios, such as financial transaction networks, communication systems, and social platforms, where new temporal graphs continuously emerge and require immediate adaptation. Consequently, this efficiency establishes MiNT as a highly practical and deployable framework for advancing the development of temporal graph foundation models.

