# OpenReview forum: "MiNT: Multi-Network Transfer Benchmark for Temporal Graph Learning"
_NeurIPS.cc/2025/Datasets_and_Benchmarks_Track — NeurIPS 2025 Datasets and Benchmarks Track poster_

### Official Review · Reviewer_x3JJ · 2025-06-21

**Rating:** 4
**Confidence:** 3

**Summary:**

This paper proposes a dataset benchmark for temporary graph learning, which contains 84 temporal networks/graphs and 8 temporal social interaction networks. The benchmark is designed for studying transferability, neural scaling, and the development of foundation models for temporal graphs. The authors also propose an approach to learn a temporary graph foundation model and the proposed method achieves promising results for both in-domain and cross-domain datasets.

**Dataset Code Accessibility:**

Yes

**Dataset Code Comments:**

The code and datasets are easily accessible.

**Ethical Considerations:**

No, there are no or only very minor ethics concerns

**Final Justification:**

The replies are detailed and address most of my concerns, and I keep my positive score.

**Limitations Weaknesses:**

1. In Figure 2, why are the training datasets packed into different sizes, and multiple MiNT models are learned? The authors should provide a clear description.
2. More baselines are needed for Table 4. Besides, why MiNT-12 is used for the cross-domain transfer study?
3. The training and testing times of both the baselines and MiNT should be provided to have a clear comparison.
4. For the last sentence of line 150, the statement is not correct, as the test graphs are also from a different domain.
5. What is the difference between existing temporary graph benchmarks and the proposed one? Can these benchmarks be used for the transferability study?

**Strengths Contributions:**

1. The temporary graph dataset benchmark is important for the research community.
2. All the datasets and codes are provided and accessible.
3. The proposed method achieves promising results for both in-domain and cross-domain datasets.

---

> ### Author Rebuttal · Authors · 2025-07-30
>
> *We thank the reviewer for their valuable feedback, which, along with comments from others, has guided significant improvements to our paper. We hope our revisions and responses merit reconsideration of a higher evaluation.*
>
> ---
> &nbsp;
> ## W1. Figure 2 sizes.
> >In Figure 2, why are the training datasets packed into different sizes, and multiple MiNT models are learned? The authors should provide a clear description.
>
> **Response:** Thank you for the question. In Figure 2, we depict the MiNT framework and highlight how we train MiNT models of varying sizes (for example, MiNT-2, MiNT-4, MiNT-8, up to MiNT-64) each corresponding to a different number of training networks. These configurations are deliberately chosen to investigate the effect of data scale on transfer performance. Each model is trained on a distinct number of networks (e.g., 2, 4, 8, 16, 32, 64) and evaluated in a zero-shot setting on the same set of held-out test networks. This design allows us to systematically analyze whether and how increasing the training set diversity and size leads to better generalization to unseen temporal graphs.
>
> We further explain this setup in Section 5.1, where we detail the training procedure, evaluation metrics, and backbone models used. Table 2 and Figure 4 empirically validate that as the training set scales, MiNT models consistently achieve stronger generalization and higher AUC scores, with MiNT-64 achieving the best average rank and win ratio.
> We will update the figure caption and narrative around Figure 2 in the final version to clarify the progression of data sizes and their purpose. Thank you again for your helpful suggestion.
>
> ---
> &nbsp;
> ## W2. Table 4 results.
> >More baselines are needed for Table 4.
>
> **Response:** Thank you for your question. We note that all baseline models, including recent state-of-the-art methods such as GraphPulse, ROLAND, EvolveGCN, and GCLSTM, are comprehensively evaluated in Table 1 (our main results).
>
> Table 4 serves a different purpose: it focuses on the impact of cross-domain transferability, examining how training MiNT on a **balanced** mixture of social and financial networks (MiNT-12 Mix) improves performance on test networks from both domains. To isolate the effect of this mixed-domain training, we compare MiNT-12 Mix to MiNT-12 (trained only on financial networks), and the best-performing (Table 1) HTGN model (trained per test dataset). Mint-12 Mix has six financial networks plus six networks from other domains, hence a balanced combination.
>
> However, we also agree that more models could help better understand the impact of the cross-domain transfer. We are currently running our code to prepare GcLSTM results until the rebuttal deadline, and we will add them to the main article.
>
> ---
> &nbsp;
>
> ## W2. Mint Mix 12.
>
> >Besides, why MiNT-12 is used for the cross-domain transfer study?
>
> **Response:** Thank you for your question. Our choice reflects a careful design to create a balanced setting. We chose MiNT-12 for the cross-domain transfer study to ensure a balanced training configuration between the two domains. Specifically, we included six social networks and matched them with six transaction networks, resulting in 12 training networks in total. This design avoids allowing the transaction domain, which contains a much larger pool of available datasets (64 networks), to dominate the learning process.
> By keeping the number of networks from each domain equal, we aimed to prevent bias toward any single domain and better assess whether MiNT can generalize across structurally and semantically different graph types. This setup allows us to isolate the impact of domain diversity during pretraining and evaluate the ability of MiNT to perform cross-domain zero-shot inference fairly.
>
> ---
> &nbsp;
>
> ## W3. Times.
> >The training and testing times of both the baselines and MiNT should be provided to have a clear comparison.?
>
> **Response:** Thank you for this valuable comment. We have prepared the table below for this question, which shows that, on average, the MiNT is 180 times more efficient than training separate models per network.
>
> One of the core strengths of MiNT is its ability to perform zero-shot inference. MiNT is trained only once on the training networks, and during evaluation on separate test networks, it performs a single forward pass without any training. This makes MiNT extremely efficient, especially in real-world scenarios where new temporal graphs frequently emerge and training a separate model for each network is time-consuming and computationally expensive.
>
> To demonstrate this efficiency, we compare the training time of a single HTGN model, trained individually on each test network, with the inference time of the MiNT model on the same networks by dividing the time needed to train a single model on that network to be prepared for inference and a pre trained MiNt inference time and we call this metric Efficiency Ratio. As shown in the table, MiNT is on average over 180 times more efficient than training a single model, while still delivering competitive or even superior performance.
> This trend is not unique to HTGN. It holds for any single model architecture, since every single model must be trained separately on each test network, whereas MiNT generalizes across networks with a single pretrained model.
>
> These results highlight the importance of transferability in temporal graph learning. MiNT enables rapid deployment on new networks without retraining, providing a scalable and practical path for future research and applications, and advancing the development of temporal graph foundation models.
>
> We will include this finding in the paper to further emphasize the value and practical advantages of transferable models.
>
> | Dataset   | HTGN (Sec) | MiNT (Sec) | Efficiency Ratio |
> |-----------|------------------------------|----------------------|------------------|
> | EVERMOON  | 196.18                       | 2.03                 | 96.64            |
> | DOGE2.0   | 392.16                       | 1.47                 | 266.78           |
> | SDEX      | 516.29                       | 3.21                 | 160.84           |
> | BAG       | 504.99                       | 4.06                 | 124.38           |
> | DINO      | 451.06                       | 4.91                 | 91.87            |
> | WOJAK     | 665.12                       | 2.67                 | 249.11           |
> | XCN       | 755.11                       | 8.33                 | 90.65            |
> | HOICHI    | 1262.99                      | 5.72                 | 220.80           |
> | Metis     | 1359.59                      | 12.93                | 105.15           |
> | QOM       | 1681.95                      | 8.34                 | 201.67           |
> | MUTE      | 1679.20                      | 13.74                | 122.21           |
> | GLM       | 1896.96                      | 15.35                | 123.58           |
> | ETH2x-FLI | 2931.61                      | 13.47                | 217.64           |
> | REPv2     | 3275.41                      | 16.70                | 196.13           |
> | DERC      | 3486.62                      | 11.49                | 303.45           |
> | BEPRO     | 3789.14                      | 15.74                | 240.73           |
> | stkAAVE   | 3194.12                      | 15.81                | 202.03           |
> | ADX       | 3673.39                      | 16.57                | 221.69           |
> | MIR       | 4525.21                      | 28.04                | 161.38           |
> | TRAC      | 6596.15                      | 29.93                | 220.39           |
> | **Average** | **2141.6625**                | **11.5255**           | **180.86**        |
>
>
> ---
> &nbsp;
>
> ## W4. Incorrect lines.
> >For the last sentence of line 150, the statement is not correct, as the test graphs are also from a different domain.
>
> **Response:** Thank you for your comment. We will update the article to address your concern and clarify the sentence in question.
>
> ---
> &nbsp;
>
> ## W5. Benchmarks.
> >What is the difference between existing temporary graph benchmarks and the proposed one? Can these benchmarks be used for the transferability study?
>
> **Response:** Thank you for the thoughtful question. While existing temporal graph benchmarks such as EdgeBank, ROLAND, and TGB have advanced evaluation for within-network tasks, they are not well-suited for studying transferability, which requires training on a large set of distinct temporal networks, which is not feasible with small-scale benchmarks composed of fewer than 10 graphs.
>
> In contrast, MiNT includes 92 temporal graphs, with 84 newly introduced real-world transaction networks. This significantly larger and more diverse benchmark enables, for the first time, systematic pretraining and evaluation of transferability across both domains and networks.  We will include this discussion in the related work.
>
> | Property                      | EdgeBank [1] | ROLAND [2] | TGB [3] | GraphPulse [4] | Ours (MiNT) |
> |----------------------------|:------------:|:----------:|:-------:|:--------------:|:-----------:|
> | # temporal graphs            |      13      |     8      |    9    |       9        |   **92**    |
> | # of newly introduced graphs |      6       |     0      |    8    |       7        |   **84**    |
>
>
>
>
> [1] Poursafaei, F. et al. (2022). Towards better evaluation for dynamic link prediction. NeurIPS.
>
> [2] You, J. et al. (2022, August). ROLAND: graph learning framework for dynamic graphs.  KDD.
>
> [3] Huang, S., et al. (2023). Temporal graph benchmark for machine learning on temporal graphs. Neurips.
>
> [4] Shamsi, K., et al. (2024, March). GraphPulse: Topological representations for temporal graph property prediction. ICLR.

---

> > ### Comment · Reviewer_x3JJ · 2025-08-03
> >
> > Thanks for the detailed reply.

---

### Official Review · Reviewer_dGZv · 2025-06-23

**Rating:** 4
**Confidence:** 5

**Summary:**

This paper studies transferability in temporal graph learning, which is a timely topic. To support this investigation, the authors introduce 84 transaction networks derived from the Ethereum blockchain. They propose a training framework that enables existing TGNNs to be trained on multiple DTDGs. A key empirical observation is that model performance improves as the number of training networks and the number of snapshots increase. Furthermore, the proposed transfer model achieves performance comparable to single models in both within-domain and cross-domain settings.

However, I view this paper more as a research track submission, primarily centered around the introduction of a new training framework.

**Additional Feedback:**

**Questions**

(1) Some of the datasets show better performance with single-model training compared to the transfer model. Do you expect the transfer model to consistently outperform single models in the future? If not, what is the practical significance of transferability?

(2) What is the compatibility of your training framework? Can it be easily integrated with a broader range of TGNN architectures?
Can this framework be extended to TGNNs operating on CTDGs?

(3) Do you believe Large Language Models (LLMs) can achieve similar transferability in temporal graph tasks? If not, what do you see as the main bottleneck?

(4) **To what extent can we trust neural scaling? Is its effectiveness limited to particular scenarios?**

**Dataset Code Accessibility:**

Yes

**Dataset Code Comments:**

The datasets are publicly available online and easily accessible for research purposes.

The codebase is well-structured, which supports the reproducibility of the experiments.

**Ethical Comments:**

The introduced datasets have already been preprocessed by the authors, which may limit access to raw data. Therefore, this lowers potential ethical concerns, as sensitive or private information is likely to have been removed during preprocessing.

**Ethical Considerations:**

No, there are no or only very minor ethics concerns

**Final Justification:**

During the deep discussion with the authors, my concerns have been addressed mostly.

The only concern is the term usage "Scaling".

**Limitations Weaknesses:**

(1) There is no specific comparison in statistics with existing DTDG datasets. This makes it difficult to assess the novelty and value of the new datasets in DTDG learning. Concretely, to show the significance and contribution of the proposed datasets in the context of DTDGs, it is essential to provide a comprehensive comparison against existing DTDG benchmarks. The paper should include a detailed quantitative and qualitative evaluation covering key dataset characteristics such as:

- **Scale:** Number of nodes, edges, and temporal snapshots.
- **Domain diversity:** The variety of application domains represented in the dataset.
- **Text attributes:** The availability of text-based node or edge attributes.
- **Vector-based features:** The inclusion of vector-based features.

In addition, the literature on discrete-time dynamic graph learning remains relatively limited compared to continuous-time approaches. To strengthen the benchmark evaluation, it would be valuable for the authors to include performance comparisons with the latest state-of-the-art methods [1-3]. This would provide a more comprehensive and fair assessment of the proposed approach and datasets.

(2) For a dataset-focused paper, more details about the datasets should be included in the main text rather than in the appendix. Additionally, such papers should ideally highlight potential research directions that the datasets enable, which are currently lacking in this work.

(3) *Minor:* It would be clearer to use the term “Multi-Graph” instead of “Multi-Network”. On page 5, the "70-15-15 split" should be revised to "70%-15%-15% split" for consistency in formatting.

**Reference:**

> [1] [Dyted: Disentangled representation learning for discrete-time dynamic graph](https://dl.acm.org/doi/abs/10.1145/3580305.3599319), KDD2023

> [2] Representation Learning of Temporal Graphs with Structural Roles, KDD’2024.

> [3] [Wingnn: Dynamic graph neural networks with random gradient aggregation window](https://dl.acm.org/doi/abs/10.1145/3580305.3599551), KDD’2023.

**Strengths Contributions:**

(1) Transferability is an important and timely topic in temporal graph learning. This paper presents a simple yet effective training framework to address it.

(2) An interesting empirical observation is that model performance improves as the number of training networks and the number of snapshots increases.

(3) While the paper is well-written, it is not well-organized for a dataset-focused submission, as much of the dataset-related content is placed in the appendix rather than the main text.

---

> ### Author Rebuttal · Authors · 2025-07-30
>
> *We thank the reviewer for their valuable feedback, which, along with comments from others, has guided significant improvements to our paper. We hope our revisions and responses merit reconsideration of a higher evaluation.*
>
> ---
> &nbsp;
> ## W1. Datasets.
> >There is no specific comparison in statistics with existing DTDG datasets.
>
> **Response:** Thank you for this comment. MiNT has the following advantages over existing datasets of TGB, ROLAND and EDGEBANK.  First, we have the largest number of newly contributed datasets (i.e., 84) to temporal graph learning, compared to a maximum of 8 in TGB. Overall, **our MiNT contains 92 datasets, whereas our competitors contain 13 at most**.  Second, **our datasets have as many as 2178 snapshots, compared to 733 in ROLAND**. Our analysis in Appendix Tables 5, 6, and Figure 7 shows the importance of our datasets.
>
> We provide the following comparison with the most relevant DTDG benchmarks and frameworks:
>
> | Property                          | EdgeBank [1] | ROLAND [2] | TGB [3] | GraphPulse [4] | Ours (MiNT) |
> |:---------------------------------:|:------------:|:----------:|:-------:|:---------------:|:-----------:|
> | Number of temporal graphs included |      13      |     8      |    9    |        9        |   **92**    |
> | Number of newly introduced graphs  |      6       |     0      |    8    |        7        |   **84**    |
>
>
> EdgeBank proposes a few small but challenging graphs, mostly focused on edge prediction tasks. ROLAND is a framework paper and does not introduce new temporal graphs. TGB focuses on a small number of distinct networks. GraphPulse includes 9 datasets with 7 financial networks for one year. In contrast, MiNT introduces 84 new temporal graphs as well as 8 existing social networks.
>
> [1] Poursafaei, F. et al. (2022). Towards better evaluation for dynamic link prediction. NeurIPS.
>
> [2] You, J., et al. (2022, August). ROLAND: graph learning framework for dynamic graphs. KDD.
>
> [3] Huang, S., et al. (2023). Temporal graph benchmark for machine learning on temporal graphs. NeurIPS.
>
> [4] Shamsi, K., et al. (2024, March). GraphPulse: Topological representations for temporal graph property prediction. ICLR.
>
>
> ---
> &nbsp;
> ## W2. SOTA Models.
> >No comparison is made with recent SOTA DTDGs... ..., performance comparisons with the latest state-of-the-art methods [1-3].
>
> **Response:** Thank you for your helpful comment and for suggesting recent sota methods. With your suggestion, we have included WinGNN [3] as a new baseline. We adapted the model to our experimental setting and trained it on the same set of test networks used in our benchmark.
>
> Our evaluation shows that MiNT-64 outperforms WinGNN in 15/20 test networks based on AUC, despite MiNT-64 requiring no training on the test networks, whereas WinGNN is trained specifically on each network. This highlights the strength and transferability of MiNT and demonstrates the value of our benchmark. We will include WinGNN as an additional baseline in the final version of the paper.
>
> | Dataset   | MiNT-64 (AUC) | WinGNN (AUC) | MiNT-64 (AP) | WinGNN (AP) |
> |-----------|----------------|----------------|----------------|----------------|
> | WOJAK     | **0.524±0.027** | 0.511±0.026 | **0.837±0.019** | 0.827±0.020 |
> | DOGE2.0   | 0.538±0.038 | **0.577±0.038** | 0.921±0.014 | **0.931±0.007** |
> | EVERMOON  | 0.517±0.039 | **0.525±0.114** | 0.551±0.021 | **0.577±0.071** |
> | QOM       | **0.647±0.019** | 0.645±0.099 | 0.635±0.027 | **0.683±0.117** |
> | SDEX      | 0.614±0.020 | **0.726±0.000** | 0.699±0.021 | **0.732±0.003** |
> | ETH2x-FLI | **0.729±0.015** | 0.617±0.056 | **0.784±0.007** | 0.624±0.093 |
> | BEPRO     | **0.782±0.003** | 0.736±0.018 | **0.816±0.014** | 0.743±0.044 |
> | XCN       | **0.851±0.043** | 0.586±0.029 | **0.861±0.042**| 0.577±0.037 |
> | BAG       | **0.931±0.028** | 0.485±0.105 | **0.889±0.043** | 0.320±0.041 |
> | TRAC      | **0.785±0.008** | 0.752±0.007 | **0.758±0.021** | 0.726±0.007 |
> | DERC      | **0.798±0.027** | 0.674±0.044 | **0.741±0.024** | 0.562±0.060 |
> | Metis     | **0.760±0.025** | 0.690±0.039 | **0.639±0.077** | 0.547±0.064 |
> | REPv2     | **0.789±0.020** | 0.744±0.026 | **0.729±0.011** | 0.688±0.046 |
> | DINO      | **0.779±0.113** | 0.628±0.251 | **0.765±0.119** | 0.681±0.194 |
> | HOICHI    | 0.765±0.018 | **0.769±0.101** | 0.594±0.012 | **0.621±0.072** |
> | MUTE      | **0.673±0.013** | 0.593±0.054 | **0.647±0.048** | 0.523±0.069 |
> | GLM       | **0.831±0.024** | 0.530±0.004 | **0.838±0.032** | 0.500±0.002 |
> | MIR       | **0.836±0.016** | 0.742±0.015 | **0.811±0.019** | 0.744±0.009 |
> | stkAAVE   | **0.709±0.022** | 0.572±0.018 | **0.759±0.019** | 0.632±0.016 |
> | ADX       | 0.679±0.024 | **0.733±0.023** | 0.628±0.012 | **0.678±0.045** |
>
>
> ---
> &nbsp;
> ## W3. Dataset details.
> >While the paper is well-written, it is not well-organized for a dataset-focused submission, as much of the dataset-related content is placed in the appendix rather than the main text.
>
> **Response:** Thank you for the thoughtful feedback. Since our paper introduces a novel task that studies transferability across temporal graphs, we prioritized presenting the empirical results and key findings in the main text due to space limitations. Nonetheless, Fig 3 of the main article gives comprehensive dataset descriptions. We agree that dataset details should be more visible in a dataset-focused paper. In the revised version, we will move the lengthy dataset descriptions (Appendix D.2) into the main text.
>
>
> ---
> &nbsp;
> ## W4. New directions.
> >Such papers should ideally highlight potential research directions that the datasets enable, which are currently lacking in this work.
>
> **Response:** Thank you for this insightful comment. Our paper introduces the novel task of temporal transferability, which opens a new research path aimed at building large temporal graph foundation models, as mentioned in lines 361-363. Our large benchmark of 84 networks, along with 8 social interaction networks, allows studying both in-domain and cross-domain transfer, as well as testing various aspects of GNNs such as temporal pooling. These directions have the potential to reshape temporal graph learning, moving toward more general-purpose, transferable models.
>
> We appreciate the suggestion and will revise the discussion in Section 5 to more clearly highlight these future research directions.
>
> ---
> &nbsp;
> ## W5. Terminology.
> >It would be clearer to use the term “Multi-Graph” instead of “Multi-Network”. On page 5, the '70-15-15 split' should be revised to '70%-15%-15% split' for consistency in formatting.
>
> **Response:** Thank you for your comment. We used the term "network" for N as it rhymes with "MiNTing" new data assets. We will update the paper formatting to address your split concern.
>
> ---
> &nbsp;
> ## W6. Single performance.
> >.. better performance with single-model training compared to the transfer model. Do you expect the transfer model to consistently outperform...? If not, what is the practical significance of transferability?
>
> **Response:** Thank you for raising this important point. We would like to emphasize that the single models are trained directly on the tested networks, while MiNT models are never trained on these test networks.  Despite this, MiNT achieves comparable or even better results in many cases, demonstrating strong generalization capabilities. This is particularly significant in real-world scenarios like ERC20 token networks, where thousands of new graphs emerge frequently, and retraining a specialized model for each new network is costly. This is the main significance of transferability.
>
> Large vs dedicated model perf. is still a hot research area in many domains (e.g., LLMs and CV models). While we do not claim that a transfer model will always outperform a dedicated single-model (but MiNT results are supporting evidence for large models), its strength lies in scalability, efficiency, and immediate applicability without retraining. Future work can further improve the architecture and training strategies. Importantly, the benchmark lays the foundation for exploring such directions, something that was not previously possible.
>
> ---
> &nbsp;
> ## W7. CTDG.
> >... Can this framework be extended to TGNNs operating on CTDGs?
>
> **Response:** Thank you for raising this point. Our framework can be adapted to work with CTDG methods; the same steps of order shuffling and context switching would apply. This is because CTDG methods also use time-evolving node embeddings, which have to be reset when switching to a new network. In addition, we expect order shuffling to be equally as important for CTDG methods as it helps achieve more IID training across graphs. The main changes needed would be to work with event streams and require the intensive step of temporal neighbor sampling, as many TGNNs require. The main limiting factor is that we do not have such large-scale, continuous data. Therefore, we leave it as promising future work.
>
> ---
> &nbsp;
> ## W8. LLMs and Neural Scaling.
> >... Can LLMs achieve transferability?... Neural Scaling?
>
> **Response:** Thank you for this insightful question. Recent studies by our team show that LLMs can perform link prediction, yet they remain fundamentally limited by context window size[1]. Only small subgraphs can be textualized, making large graph-level tasks difficult. In contrast, TGNNs can handle millions of edges efficiently with far fewer parameters.
>
> **Scaling.** We trained and tested the model also on social networks, and observed that MiNT training also benefits from the inclusion of the additional domain as shown in Appendix I. We also evaluated MiNT in a mixed domain setting by training on 12 networks from two separate domains. In this case, shown in Table 4, MiNT achieved the best average rank. These results indicate that the model benefits from seeing a larger number of networks during training.
>
>
>
> [1] Are Large Language Models Good Temporal Graph Learners? https://arxiv.org/pdf/2506.05393

---

> > ### Comment · Reviewer_dGZv · 2025-08-01
> > **Neural Scaling Concern**
> >
> > Thank you for your detailed reply. I still have some concerns. After closely examining the social network datasets, I cannot discern a consistent scaling trend. Could you clarify why this pattern emerges clearly in the MiNT datasets but not in the social network data? Does this mean the neural scaling is not universal and may be specific to MiNT?
> >
> > Another point: why does MiNT show such a clear trend? Is it because of particular dataset characteristics, the smaller snapshot size, or other statistical factors? In my view, the term “scaling” should be used more cautiously if the effect appears limited to this single case.

---

> > > ### Author Response · Authors · 2025-08-02
> > > **Thank you for your thoughtful feedback. We provide the following clarifications to address your concerns.**
> > >
> > > >  ... Could you clarify why this pattern emerges clearly in the MiNT datasets but not in the social network data? Does this mean the neural scaling is not universal and may be specific to MiNT?
> > >
> > > **Response:**  Neural scaling is not specific to MiNT’s token networks, the trend holds across social networks.  While the strongest scaling pattern is observed in MiNT’s token networks, it is not limited to this domain. In Appendix I, Table 7, we demonstrate that MiNT also exhibits consistent improvement when trained on a growing number of social networks, outperforming a single-model baseline in one of the datasets with the biggest MiNT model. This indicates that neural scaling emerges in heterogeneous domains as well, provided sufficient data is available. The results support the broader hypothesis that model performance improves with training scale.
> > > To further evaluate the consistency of this trend, we conducted an additional snapshot-level scaling experiment using MiNT-64. We trained the model on different amounts of historical data (snapshots) taken directly before the test split from each dataset. As shown below, model performance improved as more snapshots were used, reinforcing that neural scaling also applies across the temporal dimension:
> > > #### Table 1. Average test ROC AUC scores on 20 unseen token networks when MiNT-64 is trained using varying numbers of snapshots per training dataset.
> > > | Snapshots for Train (Per train dataset) | MiNT-64 Average Test AUC (on Unseen networks) |
> > > |-----------|--------|
> > > | 50        | 0.691 |
> > > | 100       | 0.681 |
> > > | 200       | 0.702 |
> > > | 500       | 0.712 |
> > > | Full      | 0.727 |
> > >
> > > These findings, which will be added to the final version of the paper, provide further evidence that MiNT supports multi-dimensional scaling and transfer learning analysis.
> > >
> > > ---
> > > >  … why does MiNT show such a clear trend? Is it because of particular dataset characteristics, the smaller snapshot size, or other statistical factors? In my view, the term “scaling” should be used more cautiously if the effect appears limited to this single case.
> > >
> > > **Response:** The clearer trend in MiNT’s financial networks is a strength, and enables robust study of transfer learning in temporal graph learning. MiNT’s token networks are constructed from real Ethereum transaction data. Although each token network evolves independently, they are all financial networks where edges represent financial transfers, nodes represent investors, and temporal patterns reflect market dynamics (e.g., bursty activity, adoption phases, market collapse, etc.). This domain consistency leads to stronger signal alignment across networks and allows MiNT to expose transferable patterns more clearly. The resulting scaling behavior is a positive outcome, demonstrating that the benchmark is structured in a way that enables meaningful generalization studies.
> > >
> > > In contrast, social networks vary substantially in interaction patterns and structure. For instance, RedditB reflects content-based engagement, Email-Eu represents organizational communications, and MathOverflow models Q&A exchanges. These networks differ in edge semantics, activity patterns, and temporal resolution, making it harder to extract a unified signal. Still, MiNT is able to show positive scaling in this diverse domain, confirming the flexibility and strength of the framework and the possibility of temporal transfer learning even over these networks.
> > >
> > > We intentionally evaluated MiNT on social networks to demonstrate that the task of temporal graph property prediction is not exclusive to financial data. By expanding to other domains, we highlight the general utility and potential of this task and underscore the need for new models to match or exceed MiNT’s strong baseline. Our goal is for MiNT to serve not just as a dataset, but as a benchmark and leaderboard to guide progress in the field. This will show the possibility of this research direction and motivate researchers to investigate this domain for robust models. In the paper, we use the term “neural scaling” following established definitions referring to empirical performance gains as training data increases.

---

> > > > ### Comment · Reviewer_dGZv · 2025-08-06
> > > >
> > > > Thank you for your response. I have carefully reviewed your comments and would like to adjust my score to be more positive.
> > > >
> > > > For the purposes of publication, I recommend that the authors use the term "scaling" with caution, as it can easily be confused with the widely discussed concept of "Scaling Laws." I suggest replacing it with a more precise term to avoid potential misunderstanding.

---

> > > > > ### Author Response · Authors · 2025-08-08
> > > > >
> > > > > Thank you very much for your consideration. We will follow your advice and rewrite the scaling discussion with better-defined terms.

---

### Official Review · Reviewer_45vc · 2025-07-04

**Rating:** 5
**Confidence:** 5

**Summary:**

The authors of this work propose a benchmark to study transfer learning in temporal graphs. They introduce a data set of 84 blockchain transaction networks and propose a multi-network training procedure, which extends the traning process to multiple temporal graphs, while resetting the latent space representation for each graph in the training set. Addressing the task of predicting the growth/shrinkage of a network (in terms of the number of edges) in a future time window, they first train six TGNN models on individual networks and report the performance. They compare this to the performance of two TGNN architectures (GC-LSTM and HTGN) which are trained on 64 networks and evaluated on 20 test networks from the same domain. The authors further consider a mixed-domain scenario, where the models are trained on a mix of blockchain transaction networks and temporal social networks, studying zero-shot performance in unseen networks from both domains. The results of the study indicate a neural scaling law for temporal graph learning, where the zero-shot performance increases with the number of networks used in the training process.

**Additional Feedback:**

I would like to thank the authors for this well-written and interesting work, which I think addresses an important open problem in temporal graph learning and presents interesting results. While reviewing the article, I came accross the questions below, which I invite the authors to discuss.

The task of predicting whether the network grows or shrinks appears to be very peculiar and not motivated well from a practical point of view. I would expect these networks to exhibit strong long-term trends of growth or shrinkage, which could make the prediction rather simple since the network will grow or shrink in many consecutive snapshots. Also, the definition of growth and shrinkage is defined in terms of the number of edges rather than the number of nodes, which seems unintuitive. Why did the authors not design their benchmark across a more challenging task, such as node property prediction or link prediction? Also, there are numerous graph-level metrics for temporal graphs that could easily be calculated and serve as graph-level property to predict. I consider this the most important issue of this work, as it raises the question whether the observed transferability of models is due to the simplicity of the task.

While I appreciate that the authors provide a new data set with 84 blockchain transaction networks that are used for training, I found this choice peculiar. I expect blockchain transaction networks to contain very specific temporal patterns that may not generalize well to other networks, which could also explain the rather modest performance. I would assume that the use of temporal graphs from different domains from a large network database like Netzschleuder could help to further improve the result (possibly in future work).

One possible limitation that I see is the context switching procedure of the proposed MiNT algorithm, which seems to assume that the TGNNs uses time-varying embeddings. This holds for many but not all TGNNs, e.g. those that compute a single i.e. static embedding that represents the temporal patterns of a  node across the whole time series. Could the same procedure also be applied to such TGNNs?

Related to the previous point, I wonder about the choice of the two specific TGNNs included in the evaluation of the transfer results.  What was the rationale behind the choice of those? And how easy would it be to transfer the training algorithm to popular TGNN architectures, like TGAT, TGN, CAWN or others? Given that the authors argue that their transfer learning algorithm is agnostic of the TGNN, I specifically could not follow the decision of not including EvolveGCN, GraphPulse and ROLAND for the transfer learning setup, which would have made for an interesting comparison.

In the results of the ablation study, I found it surprising that the context switching procedure, which is presented as a crucial part of the training algorithm, has a rather small effect on the performance. I think this should be discussed better in the paper.

Finally, I think the following work should be included in the discussion of the state-of-the-art:

Sidharth Agarwal et al.: A Transfer Framework for Enhancing Temporal Graph Learning in Data-Scarce Settings, arXiv 2503.00852

Notably, this work uses a very different approach but essentially addresses a similar problem. Also, the authors of this work address a future link prediction task, which I consider a more interesting task as the growth/shrinkage prediction task studied in the present work.

**Dataset Code Accessibility:**

Yes

**Dataset Code Comments:**

Both the code and the data are available in public repositories. The usage of the code is explained well and those parts of the code that I personally checked are well-documented by explanatory comments.

**Ethical Considerations:**

No, there are no or only very minor ethics concerns

**Final Justification:**

In the rebuttal, the authors have addressed my concerns regarding the difficulty of the task (by adding a simple "persistent prediction" baseline). I also trust that in a camera-ready version, the authors will better discuss the limitations, especially regarding the limited choice of data sets and the restriction to DTDGs.

I thus raise my score to accept.

**Limitations Weaknesses:**

- The network growth task addressed in the work is, in my view, not particularly interesting and may facilitate transferability, thus raising the question whether the observed results generalize to more realistic and challenging learning tasks.

- The authors only considered two temporal GNNs, namely HTGN and GCLSTM for the transfer results, excluding widely used models like TGAT, TGN, CAWN, etc. See my comments on the assumptions in the training algorithm below.

- The training approach seems to rely on assumptions that could make it unusable for some temporal GNNs. More details in the comments below.

- Surprisingly, the results of the ablation study indicate that the effect of the context switching procedure, which is highlighted as an essential aspect of the training algorithm is rather small. See my comments below.

- The shared mean pooling layer used for the experiments is overly simple, and may limit transferability, especially to networks exhibiting hierarchical structures.

- The discussion of related work misses an important recent paper, which seems to address the same issue (see details in comment below).

**Strengths Contributions:**

- The paper address the open challenge of transfer learning across multiple temporal graphs, showing experiments on within-domain and cross-domain generalization for a graph property prediction task.

- The authors provide a data set of 84 bitcoin transaction networks that is used for the training process, including additional temporal social networks to test cross-domain generalization performance.

- The authors further consider a mixed domain setting, where the training networks are from two domains, testing the transferability of the model to both domains.

- The authors propose a multi-network training algorithm that extends the training process to a randomly shuffled sequence of temporal graphs while resetting the temporal embeddings

- The results presented in the paper suggest a neural scaling law for temporal graph learning, where the performance in a graph property prediction task grows with the number of training networks, exceeding the performance

- The paper is well-written and the approach is described in a clear and easy to follow manner

---

> ### Author Rebuttal · Authors · 2025-07-30
>
> *We thank the reviewer for their valuable feedback, which, along with comments from others, has guided significant improvements to our paper. We hope our revisions and responses merit reconsideration of a higher evaluation.*
>
> ---
>
>
> ## W1. Task Simplicity and Practical Relevance
>
> > The main task, predicting network growth/shrinkage, is overly simple and not well-motivated.
>
> **Response:** We thank the reviewer for the discussion. We respectfully argue that the task(s) are, in fact, both challenging and grounded in real world. Forecasting whether an asset network will grow is of immense practical significance; however, due to limited space in the main text, a detailed discussion on the usefulness of the task is given in Appendix F. In decentralized finance, predicting network growth enables the anticipation of rising activity (trade volume) and user engagement around an asset. This insight is highly valuable for both investors (deciding on the timing and magnitude of investment) and developers (learning about the traction of the asset). Hence, the network growth task we study is not just academically interesting but commercially critical. In Appendix F, we also include the largest component prediction task, which is important as it identifies investor groups.  We agree that additional graph metrics could be explored. In this work, our primary objective was to demonstrate transferability, so we focused on two standard and straightforward metrics that effectively support this goal.
>
>
> **Task difficulty:**  In domains like transaction networks, especially with emerging tokens, there is no historical record. For instance, a new token entering the market with limited history cannot rely on long-term patterns, and models must instead learn how structural indicators relate to future behavior. This makes the task essential for realistic, cold-start, and low-data scenarios.
>
>
> As shown in our benchmark results, even SOTA GNNs struggle with this task. Models fail to generalize/predict future trends, suggesting that understanding dynamic topological behavior remains an open challenge. This supports our claim that the task is far from trivial and reflects meaningful complexity. This is further supported by the summary below table.
>
> ### Single SOTA GNN Models (20 Networks)
>
> | **Model**      | **Mean AUC** | **Median AUC** | **Min AUC** |
> |----------------|:------------:|:--------------:|:-----------:|
> | **HTGN**       | 0.659        | 0.668          | 0.479       |
> | **GC-LSTM**    | 0.602        | 0.632          | 0.196       |
> | **EvolveGCN**  | 0.580        | 0.618          | 0.329       |
> | **GraphPulse** | 0.710        | 0.769          | 0.384       |
> | **ROLAND**     | 0.538        | 0.513          | 0.228       |
>
>
>
>
>
> ---
>
>
>
> ## W2. Limited Model Scope
>
> > Only two TGNN models are evaluated... I could not follow the decision of not including EvolveGCN, GraphPulse and ROLAND. What was the rationale behind the choice of HTGN and GcLSTM?
>
> **Response:** Thank you for raising this. We focus on DTDGs, which are necessitated by discrete blockchain blocks that are the source of our networks. Many TGNNs like TGAT, TGN, and CAWN are built for continuous-time dynamic graphs, where fine-grained edge timestamps are needed. These models process temporal edge batches and require neighbor sampling, making them difficult to adapt to the DTDG setting without significant modifications. We agree that testing MiNT with CTDG methods is a valuable direction for future work and appreciate the suggestion; however, we do not have the required continuous data.
>
> For our study, we selected HTGN, the best model in the GraphPulse benchmark, and GC-LSTM as a strong baseline. EvolveGCN and ROLAND were evaluated in single-network settings and performed worse. GraphPulse, as noted, is not a GNN. Since HTGN and GC-LSTM outperformed these alternatives, we used them as backbones in our transfer learning framework. We will add this discussion to the article.
>
> ---
>
>
>
> ## W3. Assumptions in the Training Procedure
>
> > Context switching assumes time-varying embeddings, which may limit MiNT’s generality.
>
> **Response:** We appreciate this observation. The goal of context switching is to reset learned node embeddings between different networks, similar to how RNNs reset hidden states between sequences. Even models that compute static embeddings benefit from this, as embeddings from different graphs must remain disentangled. In general, most TGNNs use time-evolving embeddings, so context switching is a crucial part of the training framework. MiNT remains compatible with such designs, and even if embeddings are static, the reset prevents cross-network contamination.
>
> ---
>
>
>
> ## W4. Weak Impact of Context Switching in Ablation Study
>
> > Ablation results show only small benefits from context switching.
>
> **Response:** Thank you. Although the numeric improvement from context switching appears modest (4–5%), in the context of zero-shot transfer across diverse graphs, such gains are meaningful. The mechanism prevents interference across structurally different networks during training, preserving embedding quality. We will better emphasize this rationale and clarify its importance in the final version.
>
> ---
>
>
>
> ## W5. Use of a Simple Mean Pooling Layer
>
> > Mean pooling may restrict transferability, especially for hierarchical networks.
>
> **Response:** We agree that pooling is a critical design choice. We used mean pooling for its stability. In response to this comment, we conducted experiments with max pooling and found that it results in an average 6% drop in AUC as shown below, indicating that mean pooling is more effective in our zero-shot setting. Nonetheless, we agree that adaptive or hierarchical pooling is worth exploring and hope our benchmark encourages such work. We will add these findings and include the new table in the final revision.
>
> | Dataset     | MiNT-4 Test AUC - MAX Pooling | MiNT-4 Test AUC - Mean Pooling |
> |-------------|-------------------------|--------------------------|
> | MIR         | **0.588**                   | 0.510                    |
> | DOGE2.0     | 0.500                   | **0.667**                    |
> | MUTE        |  **0.685**                   | 0.627                    |
> | EVERMOON    |  **0.622**                   | 0.373                    |
> | DERC        | **0.761**                   | 0.617                    |
> | ADX         | 0.692                   |  **0.708**                   |
> | HOICHI      | 0.583                   |  **0.795**                    |
> | SDEX        | 0.401                   |  **0.643**                    |
> | BAG         | 0.610                   |  **0.802**                    |
> | XCN         | 0.557                   | **0.774**                    |
> | ETH2x-FLI   |  **0.704**                   | 0.632                    |
> | stkAAVE     | 0.525                   |  **0.571**                    |
> | GLM         | 0.598                   |  **0.671**                    |
> | QOM         | 0.611                   | **0.624**                    |
> | WOJAK       | **0.611**                   | 0.556                    |
> | DINO        | 0.480                   | **0.827**                    |
> | Metis       | 0.558                   | **0.734**                    |
> | REPv2       |**0.746**                   | 0.725                    |
> | TRAC        | 0.572                   | **0.752**                    |
> | BEPRO       | **0.788**                   | 0.742                    |
> | **Average** | 0.6096            | **0.668**                |
>
>
> ---
>
>
>
> ## W6. Domain Bias in Dataset Choice
>
> > Using only blockchain networks may limit generalization.
>
> **Response:** Thank you for raising this. While the 84 blockchain networks form the core of our benchmark, we also conducted experiments on 8 social networks. Section 5.3 and Appendix I detail the Social-MiNT variant, which achieves strong zero-shot results on these networks, demonstrating cross-domain generalization. We also performed mixed-domain training and confirmed that MiNT remains robust. Regarding Netzschleuder, we found that many datasets lack timestamped edges, which are essential for temporal modeling. We will revise the text to better reflect these cross-domain experiments.
>
> ---
>
>
>
> ## W7. Missing Relevant Related Work
>
> > The paper omits Agarwal et al. (arXiv:2503.00852) on transfer learning for temporal graphs.
>
> **Response:** Thank you for this reference. Agarwal et al. transfer from a single source to a single target graph, whereas we scale training across many networks. Both are valuable contributions, and we will cite and discuss their work.
>
> ---
>
>
>
> ## Additional Clarification: Node/Link-level Tasks
>
> > Why not consider node or link property prediction?
>
> **Response:** Our choice is motivated by data type and availability. Unlike social or road networks, blockchain networks have new nodes appear in a snapshot and disappear in the next snapshots (see Appendix table 6 for this extreme behavior, where most networks have less than 5% overlap in train/test node sets), making link prediction challenging.  Furthermore, node labels do not exist in the data for node classification tasks. In contrast, graph-level properties are consistently well-defined, making them a natural choice. Our dataset enables future work on node and link prediction tasks, and we plan to support those extensions.
>
> ---

---

> > ### Comment · Reviewer_45vc · 2025-08-01
> >
> > Thank you for the responses to my questions, which have indeed helped to mitigate some of my concerns. In particular, I consider W2, W3, W4 and W7 to be adequately addressed, provided the promised revisions are included in the camera-ready version.
> >
> > I have a few remaining remarks though:
> >
> > W1: I appreciate the response and the results on SOTA-TGNN performance, but I am still not convinced about the difficulty of the task (especially since many GNNs are known to have difficulties assessing global-scale properties). How would a simple baseline predictor perform, that always simply predicts growth/shrinkage based on the dynamics in the previous time-window or batch? I think this would be helpful to assess how difficult the task actual is.
> >
> > W6: It is true that many of the networks in the netzschleuder database do not have timestamps, but a quick check still reveals that there are 23 temporal graph data sets that cover different areas and which would be a welcome addition to such a benchmark. In particular, some of those graphs would be suitable for CTDG models, which would allow to address this concern as well. This should at least be discussed in the limitations and future work.
> >
> > On another note, given that the authors specifically limit themselves to discrete-time temporal graphs, I think this limitation should be more clearly reflected in the title/abstract and conclusion of the paper. I see that there is a short comment on this in the appendix, but I believe that this limitation should be made clearer.

---

> > > ### Author Response · Authors · 2025-08-02
> > > **We thank the reviewer for the discussion. We provide the following clarifications to address your concerns.**
> > >
> > > >  ... How would a simple baseline predictor perform, that always simply predicts growth/shrinkage based on the dynamics in the previous time-window or batch?
> > >
> > > **Response:** We thank the reviewer for the helpful suggestion regarding simple baseline predictors. To evaluate the inherent difficulty of the task, we implemented a naive baseline, Persistent Forecasting, which predicts the future growth or shrinkage based solely on trends observed in recent history. Specifically, this model uses the number of transactions from the previous and current weeks to forecast the trend in the upcoming week. If the current week shows an increase over the previous week, it predicts continued growth, and similarly for shrinkage. The results of this baseline were removed due to space limitations, but can be found on the repository (code is at script/baselines/PersistenceForecast.py and results are in figs/figure4.pdf of Sep 2024). We include the performance of this model in the table below.
> > > #### Table 1. ROC AUC scores of MiNT transfer models (with HTGN and GC-LSTM backbones) and Persistent Forecasting on unseen test sets.Best AUC Scores in bold, second best _italic_.
> > > |Network|Persistent Forecasting|MiNT-64 (GC-LSTM)|MiNT-64 (HTGN)|
> > > |---|---|---|---|
> > > |WOJAK|0.378|**0.534±0.020**|_0.524±0.027_|
> > > |DOGE2.0|0.250|_0.551±0.022_|0.538±0.038|
> > > |EVERMOON|0.241|0.494±0.047|0.517±0.039|
> > > |QOM|0.334|0.618±0.004|_0.647±0.019_|
> > > |SDEX|0.423|0.723±0.002|0.614±0.020|
> > > |ETH2x-FLI|0.355|_0.697±0.009_|**0.729±0.015**|
> > > |BEPRO|0.393|0.746±0.015|_0.782±0.003_|
> > > |XCN|0.592|0.733±0.003|**0.851±0.043**|
> > > |BAG|0.792|0.529±0.023|_0.931±0.028_|
> > > |TRAC|0.400|0.742±0.004|**0.785±0.008**|
> > > |DERC|0.353|0.696±0.011|**0.798±0.027**|
> > > |Metis|0.423|0.697±0.013|_0.760±0.025_|
> > > |REPv2|0.321|0.733±0.019|_0.789±0.020_|
> > > |DINO|0.431|0.659±0.039|0.779±0.113|
> > > |HOICHI|0.374|0.847±0.005|0.765±0.018|
> > > |MUTE|0.536|0.636±0.003|_0.673±0.013_|
> > > |GLM|0.427|0.501±0.027|**0.831±0.024**|
> > > |MIR|0.327|_0.788±0.022_|**0.836±0.016**|
> > > |stkAAVE|0.426|0.650±0.028|_0.709±0.022_|
> > > |ADX|0.362|0.673±0.022|0.679±0.024|
> > >
> > > The results in the table demonstrate that the Persistent Forecasting baseline performs significantly worse, achieving AUC scores below random guesses in many cases. This reinforces the claim that the task is non-trivial and cannot be solved effectively by a simple heuristic alone. In contrast, MiNT-64 consistently achieve substantially higher AUC scores, with many exceeding 0.7 or even 0.8, highlighting their ability to capture complex temporal and structural patterns that naive baselines cannot. We will revise our manuscript to include the Persistent Forecasting results and their discussion.
> > >
> > > ---
> > > > ... many of the networks in the netzschleuder database do not have timestamps, but a quick check still reveals that there are 23 temporal graphs ... This should at least be discussed in the limitations and future work.
> > >
> > > **Response**: We thank the reviewer for pointing this out. While it is true that many networks in the Netzschleuder database lack timestamps, we appreciate the suggestion and acknowledge that there are indeed 23 temporal graph datasets that could be valuable additions to the benchmark, particularly for evaluating CTDG models.
> > > Due to the limited time available during the rebuttal phase and the potential effort required to incorporate these datasets into our current pipeline, we are unable to include them in this version of the work. However, we agree that integrating such datasets would be an interesting and meaningful direction for future research.
> > > We revised our Limitation and Future Work Section to include a discussion about the Netzschleuder database as follows:
> > >
> > > *While our current benchmark focuses on transaction networks with high-resolution temporal dynamics, we acknowledge the potential of incorporating a broader range of temporal graph datasets. Notably, the Netzschleuder database contains 23 temporal graphs spanning diverse domains that could complement our current benchmark and support the evaluation of CTDG models. Extending our benchmark to include temporal graphs from Netzschleuder represents a promising direction for future work, particularly for assessing the generalizability of models across different types of temporal networks.*
> > >
> > > ---
> > > >  ... the authors specifically limit themselves to discrete-time temporal graphs ... this limitation should be more clearly reflected in the title/abstract and conclusion of the paper ...
> > >
> > > **Response:** We thank the reviewer for raising this important point. We agree that our focus on discrete-time temporal graphs should be made more explicit throughout the paper. In response, we will revise the abstract and conclusion to clearly state this limitation. Additionally, we will move the limitations section from the appendix to the main text and expand our discussion on the implications and assumptions of working with discrete-time temporal graphs.

---

> > > > ### Comment · Reviewer_45vc · 2025-08-06
> > > >
> > > > I thank the authors for addressing my remaining concerns!

---

### Author Response · Authors · 2025-08-08

We sincerely thank all reviewers for their thoughtful feedback and engagement, and the area chair for their help in facilitating the discussion. Your insights will be invaluable as we prepare the final manuscript.

---

### Note · Authors · 2025-08-14

We thank the reviewers for their constructive feedback and valuable suggestions that strengthened this work along two main axes.

- We clarified the practical importance and difficulty of the growth or shrinkage prediction task and provided results from a baseline that confirmed its challenge.  We also compared our datasets with existing benchmarks, reported runtime efficiency, and conducted additional scaling experiments showing that MiNT generalizes without retraining on unseen networks.

- We explained our model and design choices, the focus on discrete-time dynamic graph methods, and the compatibility of our framework with other temporal GNNs and continuous-time approaches. We added new experiments, including WinGNN as a strong state-of-the-art baseline, pooling method comparisons, and cross-domain transfer studies.

These updates reinforce the novelty and impact of our contribution. MiNT provides the first large-scale benchmark and methodology for studying transferability across temporal graphs with 92 datasets, including 84 newly introduced ones. This enables scalable zero-shot evaluation and supports both in-domain and cross-domain transfer for the first time in temporal graph learning.

We believe MiNT will serve as a foundation for future research on general-purpose temporal graph models and inspire new directions in the graph learning community.

---

### Decision · Program_Chairs · 2025-09-18

**Decision:**

Accept (poster)

**Comment:**

This paper introduces MiNT, a large-scale benchmark of 92 temporal graphs (84 newly contributed) and a framework for studying transferability in temporal graph learning. Reviewers highlighted the novelty and value of the dataset, clear methodology, and strong empirical analysis showing scaling trends and zero-shot generalization. Concerns included the simplicity of the growth/shrinkage prediction task, limited evaluation on TGNN models, and overuse of “scaling” terminology. The authors provided extensive rebuttals, added baselines (WinGNN), efficiency analysis, and clarified limitations. Overall, the benchmark and resources will have community impact.